# Deep Reinforcement Learning for Equilibrium Computation in Multi-Stage Auctions and Contests

**Anonymous Authors** [1]

## Abstract

We compute equilibrium strategies in multi-stage games with continuous signal and action spaces as they are widely used in the management sciences and economics. Examples include sequential sales via auctions, multi-stage elimination contests, and Stackelberg Bertrand competitions. While such models are fundamental to game theory and its applications, equilibrium strategies are rarely known. The resulting system of non-linear differential equations is considered intractable for all but elementary models. This has been limiting progress in game theory and is a barrier to its adoption in the field. We show that Deep Reinforcement Learning and self-play can learn equilibrium bidding strategies for various multi-stage games. We find equilibrium in models that have not yet been explored analytically and new asymmetric equilibrium bid functions for established models of sequential auctions. The verification of equilibrium is challenging in such games due to the continuous signal and action spaces. We introduce a verification algorithm and prove that the error of this verifier decreases when considering Lipschitz continuous strategies with increasing levels of discretization and sample sizes.

## 1 Introduction

Auction theory studies allocation and prices on markets with self-interested participants in equilibrium. Nobel laureate William Vickrey (1961) was the first to model markets as games of incomplete information, using the Bayes-Nash equilibrium concept. An equilibrium bid function determines how much they bid based on their value draw and their knowledge of the prior distribution. Today, auction theory is arguably the best-known application of game theory (Klemperer, 2000). These models, however, are also applied to other economic problems like crowd-sourcing, procurement, R&D contests (Konrad et al., 2009; Vojnović, 2015), and oligopoly pricing (e.g., Stackelberg competition).

Despite the enormous academic attention, equilibrium strate-gies are only known for simple market models such as single-object auctions with independent value distributions. For complex scenarios like multi-object auctions, bidders with interdependent valuations, or non-quasilinear utilities, explicit equilibrium bid functions are usually unknown. These equilibrium problems form non-linear differential equations without general mathematical solution theory.

Numerical techniques for these equations are challenging and have received limited attention. Fibich & Gavish (2011) criticize the instability of standard techniques. Equilibrium learning, an alternative numerical approach, explores the equilibrium arising from a simple learning process where agents adapt and maximize their payoff based on others' actions (Fudenberg & Levine, 1998; Hart & Mas-Colell, 2003; Young, 2004). However, learning dynamics can fail to converge to Nash equilibrium and may result in cycles, di-vergence, or chaos, even in zero-sum games (Mertikopoulos et al., 2018; Bailey & Piliouras, 2018; Cheung & Piliouras, 2020).

However, certain types of games, like potential games, fea-ture learning algorithms that always converge to equilibrium (Monderer & Shapley, 1996; Fudenberg & Levine, 1998). Whether auction games are learnable remains unresolved, even for single-stage auctions (Bichler et al., 2023b). Re-search on equilibrium computation for auctions is limited (Bosshard et al., 2020; Rabinovich et al., 2013; Naroditskiy & Greenwald, 2007, july; Greenwald & Boyan, 2001; Walsh et al., 2004). Recently, neural equilibrium learning algo-rithms have been introduced for Bayesian auction games, finding equilibrium in various single-stage auction games (Bichler et al., 2021). Ex-post verification can determine if a learned strategy profile is an equilibrium, but this is costly without known analytical strategies and relies on strong assumptions (Bosshard et al., 2020; Bichler et al., 2023a).

Most studies focus on simultaneous-move, single-stage auc-tion games, while many practical mechanisms have multiple stages. Examples include sequential sales and multi-stage elimination contests. Myerson and Reny (Myerson & Reny, 2020) introduced multi-stage games with infinite signals and actions, referred to as continuous multi-stage games, which include Bayesian and stochastic games with finite horizons. Computing equilibrium in these games is challenging. For

finite, complete-information games, Nash equilibrium computation is PPAD-hard (Daskalakis et al., 2009). Finding Bayesian Nash equilibrium in simultaneous auctions is hard for PP and even an approximate equilibrium is NP-hard (Cai & Papadimitriou, 2014). Computing coarse correlated equilibria in stochastic games is also computationally hard (Daskalakis et al., 2022).

Verification of strategy profiles as equilibria is also challenging. While verifying Nash equilibrium in finite, complete-information games is in P, for Bayesian games with different agent types, it is hard for PP and remains NP-hard even with constant error (Cai & Papadimitriou, 2014). Verification techniques for games with continuous signals and actions are not readily applicable. Approximate methods for verifying equilibria in single-stage auctions with continuous actions have been proposed (Bosshard et al., 2020; Hosoya & Yu, 2022), but in dynamic games interdependencies naturally arise due to signals–such as priors and actions–revealing information in previous stages of the game upon which future actions can be based on. As a result, the techniques previously employed for verification are not applicable in our case.

We aim to find equilibrium in continuous multi-stage games via reinforcement learning algorithms. Unlike single-stage games, this requires considering the state of the game and different learning algorithms. Reinforcement learning (RL), developed for single-agent learning in discrete-time stochastic control processes, has been applied in various fields, including robot control and elevator scheduling (Sutton & Barto, 2018). Multi-agent RL has achieved super-human performance in finite games like chess, shogi, Go (Silver et al., 2016; 2018), and imperfect information games like poker (Brown & Sandholm, 2018; 2019). However, it is unknown how far learned strategies are from equilibrium in these large games.

Convergence guarantees for learning algorithms are limited to special game classes or require strong equilibrium properties (Lockhart et al., 2019; Marden, 2012; Macua et al., 2018, april; Leonardos et al., 2022, april; Giannou et al., 2022). Multi-stage games with continuous signals and actions do not fall into these categories. Current understanding of learning dynamics and equilibrium in games remains limited (Yang & Wang, 2020; Zhang et al., 2021).

### 1.1 Contributions

We study the convergence of deep reinforcement learning (DRL) algorithms to equilibrium in multi-stage games with continuous signal and action spaces. These games cover various strategic situations, but there is no general solution theory for deriving equilibrium in such cases. Sequential auctions with unit-demand bidders or restricted elimination contests are exceptions where analytical equilibrium

strategies can be derived. The equilibrium problem here forms a system of non-linear partial differential equations (PDEs). Hence, the solution methods differ from those in combinatorial but finite games, which have been the focus of multi-agent reinforcement learning. The continuous signal space requires finding an equilibrium bid function rather than a specific bid or action.

In the absence of analytical solutions for most continuous multi-stage games, numerical methods are crucial for advancing game theory and its applications. Standard numerical PDE solvers have failed. We use policy gradient methods like REINFORCE and Proximal Policy Optimization (PPO) and neural networks to approximate equilibrium bid functions. These algorithms are designed for single-agent tasks, and convergence results rely on the environment being stationary and Markovian (Sutton et al., 1999), which does not apply in multi-agent and imperfect information settings. Generally, these algorithms do not converge even in zero-sum settings (Littman, 1994; Srinivasan et al., 2018) or linear quadratic games (Mazumdar et al., 2020). Our experiments show we can find equilibrium strategies in sequential auctions, elimination contests, and Stackelberg Bertrand competitions using appropriate DRL algorithms. Additionally, DRL techniques allow scaling to larger instances. It is remarkable that DRL with self-play converges to such a strategy profile. We also analyze equilibrium in auctions with interdependent valuations and risk-averse bidders, where no analytical equilibrium strategy is known. We find new asymmetric equilibrium in well-known sequential sales models previously undiscovered.

Our second main contribution is an algorithm to verify an approximate global Nash equilibrium in continuous multi-stage games. Verification in these games is costly due to the infinite space of alternative strategies. We prove that the error of our verifier decreases with higher discretization levels and increased sample sizes. This provides a foundation for equilibrium solvers widely applicable in this significant game class.

## 2 The Model

In what follows, we introduce continuous multi-stage games with continuous signal and action spaces and reinforcement learning algorithms used.

### 2.1 Continuous Multi-Stage Games

We study games with multiple stages, where players can have uncountably many actions, such as bids in auctions. Also, agents receive signals that may be from uncountable sets, such as a continuous type (or value) space in auction theory. In a multi-stage game $\Gamma$, $N$ players plus nature interact simultaneously for $T$ stages. In each stage $t$, each

player $i$ and nature $i = 0$ receives a signal $s_{it}$ and chooses its action $a_{it}$. After $T$ stages, each agent receives a utility $u_i(a)$ based on an outcome $a$. Multi-stage auctions or contests are illustrative examples and the focus of this paper, but the game class is very general and allows to model single-stage Bayesian games, signaling games, and finite-horizon partially observable stochastic games. A strategy for player $i$ in stage $t$ is a mapping $\beta_{it} : S_{it} \to \Delta(\mathcal{A}_{it})$.

The most prominent equilibrium solution concept in non-cooperative game theory is the Nash equilibrium (NE) (Nash et al., 1950). Let $\Gamma$ be a multi-stage game, $\varepsilon \geq 0$, and $\beta^* \in \Sigma$ a strategy profile. Then $\beta^*$ is an $\varepsilon$-Nash equilibrium ($\varepsilon$-NE) if and only if for all $i \in \mathcal{N}$ and $\beta_i \in \Sigma_i$

$$\tilde{u}_i(\beta_i, \beta^*_{-i}) \leq \tilde{u}_i(\beta^*) + \varepsilon. \tag{1}$$

For a formal description of the model, we refer to Section A.

## 2.2 Learning for Equilibrium Selection

A game theoretic solution concept such as the NE usually follows the normative approach, telling the players how to act. This neglects the question of how the players would find and agree on an equilibrium (Ashlagi et al., 2006). Instead, we use learning as equilibrium refinement and analyze whether the resulting strategy profile is a $\varepsilon$-Nash equilibrium.

Consider a parametrization of the players' strategies so that for every $it \in L$, there is a set of parameters $\Theta_{it}$ and a mapping $\Theta_{it} \mapsto \Sigma_{it}$ that maps onto a strategy. We focus on policy gradient learning algorithms. Player $i$ updates the parameters $\theta_i^r$ with a learning rate $\eta_i^r$ in iteration $r$ by

$$\theta_i^{r+1} = \theta_i^r + \eta_i^r \cdot \nabla_{\theta_i} \tilde{u}_i\left(\beta_{\theta_i^r}, \beta_{\theta_{-i}^r}\right).$$

We focus on two common policy gradient algorithms, namely REINFORCE (Mohamed et al., 2020) and PPO (Schulman et al., 2017). We provide additional information on these DRL algorithms in Section B.

## 3 Evaluation Metrics

The ex-ante *utility loss* is a metric to measure the loss of an agent by not playing a best-response to the opponents' strategies $\beta$ (Srinivasan et al., 2018; Brown et al., 2019). It is given by $\tilde{\ell}_i(\beta_i, \beta_{-i}) = \sup_{\beta_i' \in \Sigma_i} \tilde{u}_i(\beta_i', \beta_{-i}) - \tilde{u}_i(\beta_i, \beta_{-i})$. A strategy profile $\beta$ is a $\varepsilon$-NE if and only if $\tilde{\ell}_i(\beta_i, \beta_{-i}) \leq \varepsilon$. In a setting with a known analytical equilibrium $\beta^* = (\beta_i^*, \beta_{-i}^*)$, we estimate closeness in utility of a strategy $\beta_i$ to $\beta_i^*$ by the utility loss in equilibrium by

$$\tilde{\ell}_i^{\text{equ}}(\beta_i) := \tilde{\ell}_i(\beta_i, \beta_{-i}^*) = \tilde{u}_i(\beta_i^*, \beta_{-i}^*) - \tilde{u}_i(\beta_i, \beta_{-i}^*). \tag{2}$$

Additionally, we evaluate convergence in strategy space with the probability-weighted root mean squared error of

$\beta_{it}$ and $\beta_{it}^*$ to approximate the weighted $L_2$ distance of these functions

$$L_2^{it}(\beta, \beta^*) = \left( \int (\beta_{it}(s_{it}) - \beta_{it}^*(s_{it}))^2 \right)^{\frac{1}{2}} dP_{it}(\cdot \mid \beta^*), \tag{3}$$

where $P_{it}(\cdot \mid \beta^*)$ denotes the probability measure over signals induced by strategy profile $\beta^*$. $L_2^{i,\text{avg}}$ denotes the mean over all stages. In general, we cannot evaluate the these metrics directly. Therefore, we estimate them via Monte-Carlo approximation.

## 4 Verification in Settings with Unknown Equilibrium

In cases where no analytical solution is known, estimating the best-response utility for the utility loss becomes necessary in order to verify whether a strategy profile is indeed an approximate NE. However, deriving such guarantees ex-post is challenging in continuous games with multiple stages. Without imposing any additional regularity on the strategy profile $\beta = (\beta_i, \beta_{-i})$, it is very hard to give any theoretical guarantees. Therefore, we limit each strategy $\beta_i$ to be from the set of Lipschitz continuous functions, that is denoted by $\Sigma_i^{\text{Lip}}$.

The search for best-responses is limited further to the space of pure Lipschitz continuous functions $\Sigma_i^{\text{Lip, p}}$. The utility loss with regard to pure Lipschitz continuous functions is denoted by

$$\tilde{\ell}_i^{\text{Lip, p}}(\beta) := \sup_{\beta_i' \in \Sigma_i^{\text{Lip, p}}} \tilde{u}_i(\beta_i', \beta_{-i}) - \tilde{u}_i(\beta_i, \beta_{-i}). \tag{4}$$

We face two challenges in estimating $\tilde{\ell}_i^{\text{Lip, p}}(\beta)$. First, one needs to search within the infinite-dimensional function space $\Sigma_i^{\text{Lip,p}}$ for a best-response. The second challenge pertains to the precise evaluation of ex-ante utility, even for a single strategy profile $\beta = (\beta_i, \beta_{-i})$. Only a precise estimate of the ex-ante utility allows for an accurate verification of equilibrium. Our verifier employs two core concepts to address these challenges.

First, we limit the search space by approximating the utility of a best response with a set of finite precision step functions denoted as $\Sigma_i^{\text{D}}$. Here, $\text{D} \in \mathbb{N}$ represents the number of steps or discretization points in both the domain and image space. This means that for a single agent, we consider strategies that only allow responses to signals with finite precision. Second, we employ Monte-Carlo estimation for the ex-ante utility. As a result, we only need to consider a finite set of strategies, but we can still prove that the estimated utility loss by our verifier $\ell_i^{\text{ver}}(\beta)$ serves as an upper bound for $\tilde{\ell}_i^{\text{Lip, p}}(\beta)$ with sufficient resources. This allows us to prove the following convergence guarantees for our verifier.

**Theorem 4.1** (informal). *For a given multi-stage game* $\Gamma = (\mathcal{N}, T, S, \mathcal{A}, p, \sigma, u)$, *strategy profile* $\beta = (\beta_i, \beta_{-i})$ *with* $\beta_i \in \Sigma_i$ *and* $\beta_{-i} \in \Sigma_{-i}^{Lip}$, *and a continuous utility function* $u_i$, *we have that with a sufficiently high precision* $D$ *and initial simulation count* $M_{IS}$ *the estimated utility loss is an upper bound for the utility loss over pure Lipschitz continuous functions, i.e.,*

$$\lim_{D \to \infty} \lim_{M_{IS} \to \infty} \ell_i^{ver}(\beta) \geq \tilde{\ell}_i^{Lip, p}(\beta) \text{ almost surely.}$$

For these guarantees, we make standard regularity assumptions on the game and the players' strategies. We also assume that the ex-post utility functions, denoted as $u_i$, are continuous. While the latter is not satisfied in auctions, Section C.2 provides experimental evidence for the estimation error becomes very small with increasing levels of discretization and increasing sample size. We refer to Section D.2 for a more detailed description of the verifier and to Section D.3 for the proof.

## 5 Experimental Results

We report results for sequential auctions in this section. For the standard independent private values model of a first- and second-price auction with risk-neutral bidders, we have an analytical solution (Milgrom & Weber, 2000; Krishna, 2009). We also learn the equilibrium for risk-averse bidders with interdependent valuations, for which the equilibrium strategy was unknown so far. The approach is suitable for continuous multi-stage games in general, and in Section C we report additional results on elimination contests, Stackelberg Bertrand competitions and new asymmetric equilibria that we found for the sequential second-price auction model.

The baseline model of sequential sales is as follows: Let $T$ be the number of homogeneous units for sale and there be $N > T$ bidders. In each stage $t$, there is exactly one unit for sale. Bidders are only interested in winning a single item, and they are privately informed of their valuation $v_i$ before the beginning of the first stage. Based on the submitted sealed-bids $a_{.t}$ in each stage, an auction mechanism calculates the allocation of the good and the price $p_{it}(a_{.t})$ for the winner. Here, we assume risk-neutrality with a utility of $v_i - p_{it}(a_t)$ for the winner and zero for the losers. The prices from previous stages are revealed. If only the winner's bid is announced and the winner drops out of the upcoming stages, there is no additional information on the remaining opponents that can be leveraged.

We evaluate the learning algorithms for the first-price mechanism and for different numbers of stages $T$. For simplicity, we set the number of bidders $N$ to $T + 1$ such that there remains competition in the final stage. The full results for PPO and REINFORCE, first-price sequential sales, and different numbers of objects $T$ can be found in Table 1. We use

*Table 1.* Learning results for sequential sales with various numbers of stages. We report the mean $L_2^{avg}$ (Equation 3) and utility loss $\ell^{equ}$ with respect to the analytical symmetric equilibrum (Equation 2), as well as the estimated utility loss $\ell^{ver}$ (Equation 23) across ten runs together with the standard deviations.

| $T$ | metric | REINFORCE | PPO |
|---|---|---|---|
| 1 | $L_2^{avg}$ | 0.0060 (0.0010) | 0.0047 (0.0009) |
| | $\ell^{equ}$ | 0.0001 (0.0002) | 0.0001 (0.0001) |
| | $\ell^{ver}$ | 0.0003 (0.0001) | 0.0003 (0.0001) |
| 2 | $L_2^{avg}$ | 0.0098 (0.0046) | 0.0054 (0.0020) |
| | $\ell^{equ}$ | 0.0002 (0.0002) | 0.0000 (0.0002) |
| | $\ell^{ver}$ | 0.0013 (0.0002) | 0.0003 (0.0003) |
| 4 | $L_2^{avg}$ | 0.0110 (0.0044) | 0.0056 (0.0034) |
| | $\ell^{equ}$ | 0.0003 (0.0003) | 0.0000 (0.0003) |
| | $\ell^{ver}$ | -0.0005 (0.0013) | -0.0016 (0.0010) |

the same set of hyperparameters for almost all settings (see subsection B.3). The utility loss is very low for all settings considered.

## 6 Conclusions

We investigate the application of DRL for continuous multi-stage games. Although such games are central to theory and in numerous real-world applications, we know equilibrium strategies for only a few restricted models. Infinite type and action spaces make the analysis of such games challenging and different from combinatorial games such as Go and Poker, as they have been the focus of multi-agent reinforcement learning so far. Besides, scholars are interested in equilibrium strategies of respective games. Computing such equilibrium strategies has long been considered intractable.

We employed deep reinforcement learning to approximate the equilibrium bidding strategies in these continuous multi-stage games. Interestingly, our experiments showed that these methods find equilibrium strategies in sequential auctions and contests under very different model assumptions. Importantly, the analysis can be performed quickly. We can explore new environments but also unravel new equilibria that were not known before.

A key contribution is a verifier, which is able to certify an approximate equilibrium in continuous multi-stage games, even when no analytical equilibrium is known, and prove that we receive an upper bound on the utility loss as the number of samples (games played) grows large and the level of discretization increases. The fact that deep reinforcement learning converges to an equilibrium in continuous multi-stage auctions and contests is remarkable and provides the foundation for widely applicable equilibrium solvers.

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

# A The formal game model

We assume all functions to be measurable over suitable sigma-algebras and denote with $\Delta(X)$ the set of countably additive probability measures on the measurable subsets of $X$. For measurable spaces $X$ and $Y$, a mapping $f : Y \to \Delta(X)$ is called a transition probability if, for every measurable subset $A \subset X$, the function $f(A \mid \cdot) : Y \to \mathbb{R}$ is measurable.

**Definition A.1** (Multi-stage game (Myerson & Reny, 2020)). A multi-stage game is specified by a tuple $\Gamma = (\mathcal{N}, T, S, \mathcal{A}, p, \sigma, u)$, where

1. $\mathcal{N} = \{1, \ldots, N\}$ is the set of $N \in \mathbb{N}$ players. Let $\mathcal{N}^* = \mathcal{N} \cup \{0\}$, where $0$ is nature.

2. $T \in \mathbb{N}$ is the number of stages. Let $L = \mathcal{N} \times \{1, \ldots, T\}$ denote the set of dated players and $L^* = \mathcal{N}^* \times \{1, \ldots, T\}$. For simplicity, we write "$it$" for $(i, t)$ and $t \in T$ to express $t \in \{1, \ldots, T\}$.

3. $S = \bigtimes_{it \in L} S_{it}$, where $S_{it}$ is the set of possible signals that player $i$ can receive at stage $t$. It holds $S_{i1} = \{\emptyset\}$ for all $i \in \mathcal{N}$.

4. $\mathcal{A}_{it}$ denotes the set of player $i$'s actions in stage $t$. $\mathcal{A}_{0t}$ is nature's set of actions in stage $t$. $\mathcal{A} = \bigtimes_{it \in L^*} \mathcal{A}_{it}$ denotes the set of possible *outcomes* of the game. Each element in $\mathcal{A}$ describes a game's complete roll-out. We denote the projection onto stages before $t$ by a subscript $< t$. For example, for $a \in \mathcal{A}$, $a_{<t} = (a_{ir})_{i \in \mathcal{N}^*, r < t}$ denotes the history proceeding stage $t$.

5. $p = (p_1, \ldots, p_T)$ is nature's fixed probability function, where $p_t : \mathcal{A}_{<t} \to \Delta(\mathcal{A}_{0t})$ for all $t \in T$.

6. $\sigma_{it} : \mathcal{A}_{<t} \to S_{it}$ denotes player $i$'s signal function for stage $t$, and $\sigma = (\sigma_{it})_{it \in L}$.

7. $u_i : \mathcal{A} \to \mathbb{R}$ is player $i$'s bounded utility function, and $u = (u_i)_{i \in \mathcal{N}}$.

In subsection D.1, we discuss additional assumptions such as perfect recall of players about their own actions and information received and standard assumptions about the bid functions to be pure and Lipschitz continuous, which we use in our analysis. One can model various different settings as a multi-stage game, including single-stage Bayesian games, signaling games, finitely-repeated games, and finite-horizon stochastic games. Most importantly, this definition allows for high-dimensional continuous signals and actions. By including dummy actions and signals, one can also model sequential move games or, more generally, games where a subset of players acts in each stage. For example, in the settings considered in section 5, nature moves first and draws players' types. The players receive their types as signals in the second stage and act. This can be modeled by setting $\mathcal{A}_{i1}$ to be a singleton for $i \in \mathcal{N}$. So, a two-stage sequential auction can be modeled by a three-stage game.

A strategy for player $i$ in stage $t$ is a transition probability $\beta_{it} : S_{it} \to \Delta(\mathcal{A}_{it})$. Let $\Sigma_{it}$ denote $i$'s set of strategies at time $t$ and $\Sigma_i = \bigtimes_{t \in T} \Sigma_{it}$ player $i$'s set of strategies. Finally, $\Sigma = \bigtimes_{it \in L} \Sigma_{it}$ is the set of all strategies. We denote with $\beta_{\cdot t} = (\beta_{it})_{i \in \mathcal{N}}$ the strategy vector of stage $t$. Likewise, the $\cdot t$ notation denotes corresponding product spaces and vectors for stage $t$, e.g., $\mathcal{A}_{\cdot t} = \bigtimes_{i \in \mathcal{N}^*} \mathcal{A}_{it}$ and $\Sigma_{\cdot t} = \bigtimes_{i \in \mathcal{N}} \Sigma_{it}$. We use $\tilde{\cdot}$ to refer to variables in the ex-ante state of the game. For example, $\tilde{u} : \Sigma \to \mathbb{R}$ describes the ex-ante utility with strategy profile $\beta$.

The most prominent equilibrium solution concept in non-cooperative game theory is the Nash equilibrium (NE) (Nash et al., 1950). Informally, it is a fixed point in strategy space where no player unilaterally wants to deviate from.

**Definition A.2** ($\varepsilon$-Nash equilibrium). Let $\Gamma = (\mathcal{N}, T, S, \mathcal{A}, p, \sigma, u)$ be a multi-stage game, $\varepsilon \geq 0$, and $\beta^* \in \Sigma$ a strategy profile. Then $\beta^*$ is an $\varepsilon$-Nash equilibrium ($\varepsilon$-NE) if and only if for all $i \in \mathcal{N}$ and $\beta_i \in \Sigma_i$

$$\tilde{u}_i(\beta_i, \beta^*_{-i}) \leq \tilde{u}_i(\beta^*) + \varepsilon. \tag{5}$$

We denote $\beta^*$ simply as Nash equilibrium for $\varepsilon = 0$.

The NE is prevalent for normal-form games: single-stage games with complete information. However, in games with multiple stages, one might want to exclude Nash equilibria that rely on non-credible threats or non-best responses in some subgames. One way to exclude these unwanted equilibria is to demand the strategies to be *sequentially rational*. Krishna (2009) defines this recursively as equilibria with the property that following an outcome of the current stage, the strategies in the next stage form an equilibrium. In the complete information case, this leads to *subgame perfect (Nash) equilibrium*, eliminating unreasonable Nash equilibria.

# B  Deep Reinforcement Learning Methods

Deep reinforcement learning (DRL) combines deep learning techniques with reinforcement learning algorithms to train agents capable of making decisions in complex environments. DRL algorithms learn stochastic policies (or mixed strategies in games) by interacting with the environment, receiving feedback in the form of rewards or penalties, and adjusting their behavior to maximize cumulative rewards over time. REINFORCE and Proximal Policy Optimization are two of the most important representatives.

## B.1  REINFORCE

REINFORCE is a policy gradient algorithm based on the idea of optimizing the policy directly without explicitly estimating the value function. The REINFORCE algorithm computes the gradient of the expected return with respect to the policy parameters and updates the policy accordingly. This allows the agent to learn to take actions that lead to higher rewards. This estimator is also known as the score-function estimator. REINFORCE learns mixed or distributional strategies, which gets around the problem with discontinuous ex-post utility functions discussed in Bichler et al. (2021). A player wants to maximize his or her expected utility over the opponents. The gradient is given as

$$
\begin{aligned}
\nabla_{\theta_i}\, \tilde{u}_i\left(\beta_{\theta_i}, \beta_{\theta_{-i}}\right) &= \nabla_{\theta_i}\, \mathbb{E}_{a \sim P\left(\cdot \,\mid\, \beta_{\theta_i}, \beta_{\theta_{-i}}\right)} \left[u_i(a)\right] \\
&= \nabla_{\theta_i} \int_{\mathcal{A}} u_i(a) \rho\left(a \,\mid\, \beta_{\theta_i}, \beta_{\theta_{-i}}\right) da \\
&= \int_{\mathcal{A}} u_i(a) \nabla_{\theta_i} \log\left(\rho\left(a \,\mid\, \beta_{\theta_i}, \beta_{\theta_{-i}}\right)\right) da \\
&= \mathbb{E}_{a \sim P\left(\cdot \,\mid\, \beta_{\theta_i}, \beta_{\theta_{-i}}\right)} \left[u_i(a) \nabla_{\theta_i} \log\left(\rho\left(a \,\mid\, \beta_{\theta_i}, \beta_{\theta_{-i}}\right)\right)\right],
\end{aligned}
$$

where $\rho\left(\cdot \,\mid\, \beta_{\theta_i}, \beta_{\theta_{-i}}\right)$ denotes the density function of $P\left(\cdot \,\mid\, \beta_{\theta_i}, \beta_{\theta_{-i}}\right)$. This derivation follows the policy gradient theorem (Sutton & Barto, 2018). We parametrize the neural network to output the mean and standard deviation of a Gaussian distribution so that one can assume $\rho$ to exist. Now one is able to state this expression as an expectation that can be approximated by sampling when following the well-known score-function reformulation. More details on this estimator, including a broader discussion, can be found in the study of Mohamed et al. (2020).

## B.2  Proximal Policy Optimization

Proximal Policy Optimization (PPO) has been introduced by Schulman et al. (2017) and can be considered an extension of the REINFORCE algorithm. PPO is designed to strike a balance between sample efficiency and stability during training. It addresses the challenges of policy optimization by using multiple iterations of stochastic gradient ascent, where the update in each iteration is limited to a certain range, thus avoiding large policy updates that could disrupt learning. This constraint helps maintain stability and prevents the agent from deviating too far from its previous policy.

PPO has been widely adopted and has demonstrated strong performance across a range of complex tasks. It falls in the broader class of actor-critic algorithms – thereby introducing a second network that estimates the state's value – and additionally uses a technique called trust region policy optimization, which helps to prevent the algorithm from making large, potentially harmful changes to the policy. This tends to make learning more stable compared to REINFORCE. PPO is considered a state-of-the-art method for reinforcement learning, particularly in complex environments with high-dimensional state spaces, and it has been particularly successful in combinatorial games (Yu et al., 2022).

## B.3  Hyperparameters

We employ common hyperparameters for our experiments, utilizing fully connected neural networks with two hidden layers, each consisting of $64$ nodes, and employing SeLU activations on the inner nodes (Klambauer et al., 2017). The weights and biases of these networks determine the parameters $\theta_i$. All experiments were conducted on a single Nvidia GeForce 2080Ti GPU with 11 gigabytes of RAM, accommodating a parallel simulation of $20,000$ environments. We employed the ADAM optimizer with a learning rate of $8 \times 10^{-6}$ for all experiments, except for the Stackelberg Bertrand competition, where we used a learning rate of $5 \times 10^{-5}$. The initial log-standard deviation is set to $-3.0$, except for the REINFORCE method in the elimination contest, specifically in the information case of published bids, where we set it to $-2.0$. All remaining parameters are set to the default values used in the framework by Raffin et al. (2021).

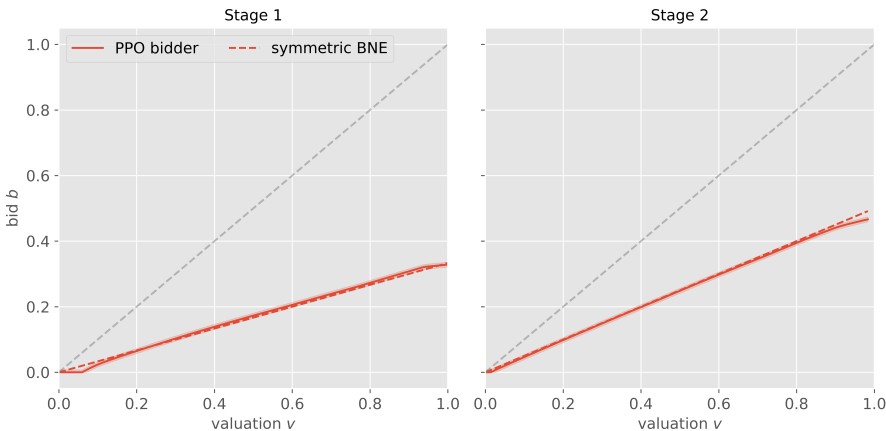

*Figure 1.* Equilibrium and PPO-based learned strategies in sequential sales with a first-price mechanism, two-stages, and three bidders.

For our verification procedure, we employ a discretization parameter of $D = 64$ and an initial number of simulations $M_{\text{IS}} = 2^{21}$ as default. We increase the discretization to $D = 128$ in the Stackelberg Bertrand competition, and reduce it to $D = 16$ for the four-stage Sequential Auction, so that the information set tree fits onto a single GPU.

## C  Additional Empirical Results

In this section, we give additional empirical results. First, we state the known analytical equilibrium strategies in the standard sequential auctions model and report our results for the first- and second-price rule. Second, we empirically analyze the number of samples needed to achieve a small approximated utility loss for the verifier in a two-stage sequential auction. Third, we report approximate equilibria in sequential auctions under risk-averse bidders, budget constraints, and interdependent priors. Additionally, we study an elimination contest and a Stackelberg Bertrand competition.

### C.1  Sequential Sales: Independent Private Values

The analytical equilibrium in the symmetric sequential auction model with independent private values and risk-neutral bidders, is as follows:

**Proposition C.1** (Krishna (2009)). *Suppose bidders have unit-demand drawn uniformly from $[0, 1]$ and $T$ units are sold by means of sequential auctions. Then, the following strategies constitute symmetric equilibria in the $t$-th stage:*

*1. First-price:*

$$\beta_{it}(v_i) = \frac{N - T}{N - t + 1} v_i,$$

*2. Second-price:*

$$\beta_{it}(v_i) = \frac{N - T}{N - t} v_i.$$

Because the ratio of supply to demand is decreasing over time, bidders are forced to increase their bids in both mechanisms. Commonly, one assumes that the prices from previous stages are revealed.

Here, we evaluate the learning algorithms for the first- and second-price mechanism and for different numbers of stages $T$. Figure 1 depicts exemplary strategies in a two-stage auction. The full results for PPO and REINFORCE, first- and second-price sequential sales, and different numbers of objects $T$ can be found in Table 2.

### C.2  Discretization and Number of Samples for the verifier

The utility loss estimates calculated by the verifier as described in Theorem D.4 depend on the level of discretization and the number of simulated games. Former ensures that game state, observation, and action space closely resemble the original

*Table 2.* Learning results for sequential sales with various numbers of stages. We report the mean $L_2^{\mathrm{avg}}$ (Equation 3) and utility loss $\ell^{\mathrm{equ}}$ with respect to the analytical symmetric equilibirum (Equation 2), as well as the estimated utility loss $\ell^{\mathrm{ver}}$ (Equation 23) across ten runs together with the standard deviations.

| mechanism | $T$ | metric | REINFORCE | PPO |
|---|---|---|---|---|
| first | 1 | $L_2^{\mathrm{avg}}$ | 0.0060 (0.0010) | 0.0047 (0.0009) |
| | | $\ell^{\mathrm{equ}}$ | 0.0001 (0.0002) | 0.0001 (0.0001) |
| | | $\ell^{\mathrm{ver}}$ | 0.0003 (0.0001) | 0.0003 (0.0001) |
| | 2 | $L_2^{\mathrm{avg}}$ | 0.0098 (0.0046) | 0.0054 (0.0020) |
| | | $\ell^{\mathrm{equ}}$ | 0.0002 (0.0002) | 0.0000 (0.0002) |
| | | $\ell^{\mathrm{ver}}$ | 0.0013 (0.0002) | 0.0003 (0.0003) |
| | 4 | $L_2^{\mathrm{avg}}$ | 0.0110 (0.0044) | 0.0056 (0.0034) |
| | | $\ell^{\mathrm{equ}}$ | 0.0003 (0.0003) | 0.0000 (0.0003) |
| | | $\ell^{\mathrm{ver}}$ | -0.0005 (0.0013) | -0.0016 (0.0010) |
| second | 1 | $L_2^{\mathrm{avg}}$ | 0.0091 (0.0007) | 0.0063 (0.0022) |
| | | $\ell^{\mathrm{equ}}$ | 0.0001 (0.0002) | 0.0002 (0.0001) |
| | | $\ell^{\mathrm{ver}}$ | 0.0000 (0.0000) | 0.0000 (0.0000) |
| | 2 | $L_2^{\mathrm{avg}}$ | 0.0075 (0.0028) | 0.0068 (0.0028) |
| | | $\ell^{\mathrm{equ}}$ | 0.0001 (0.0002) | -0.0001 (0.0003) |
| | | $\ell^{\mathrm{ver}}$ | 0.0033 (0.0005) | 0.0020 (0.0006) |
| | 4 | $L_2^{\mathrm{avg}}$ | 0.0140 (0.0036) | 0.0072 (0.0031) |
| | | $\ell^{\mathrm{equ}}$ | 0.0002 (0.0002) | 0.0000 (0.0004) |
| | | $\ell^{\mathrm{ver}}$ | 0.0050 (0.0003) | 0.0039 (0.0008) |

continuous game, whereas the latter ensures that the approximated utilities are close to their expectations (with respect to the distribution of the valuations and actions). We illustrate which precision can be reached for multiple configurations of these parameters. For this, we deploy the verifier in the two-stage sequential sales with three participants and a first-price payment rule. The opponents play their equilibrium strategy, ensuring that the exact utility loss is zero. Figure 2 shows how the discretization size and the number of simulations influence the utility loss. For a low number of initial simulations, the simulation error $\varepsilon_{M_{\mathrm{IS}}}$ dominates and strongly overestimates the utility loss. That is because our procedure chooses the maximum attainable utility over the simulated data. For a sufficient amount of simulations, the estimated utility loss becomes negative, showing the discretization error's $\varepsilon_{\mathrm{D}}$ effect. For a finer discretization, $\varepsilon_{\mathrm{D}}$ tends towards zero. We use a discretization of D = 64 and an initial number of simulations $M_{\mathrm{IS}} = 2^{21}$ if not stated otherwise.

The verifier does make use of a vectorized implementation, which allows parallel evaluation of thousands of games. The run times increase linearly as parallel batches are split into sequentially evaluated mini-batches.

### C.3 Sequential Sales: Interdependence, Risk-Aversion, and Budget Constraints

We present a series of additional experiments that explore the effects of various adaptations to the standard sequential auction setting. These adaptations aim to capture important behavioral factors and practical considerations.

One popular behavioral effect we investigate is the influence of risk aversion on bidding behavior. Bidders may exhibit risk aversion leading to diminishing marginal utilities, making it more important for them to secure a win than to maximize their payoff. To incorporate this aspect, we introduce a risk parameter, denoted as $\rho \in (0, 1]$, that modifies the utility function to $u_i(v_i, a) = (x_i(a)v_i - p_i(a))^\rho$.

Moreover, we consider the effect of budget constraints among bidders. While this assumption likely holds in virtually all practical settings, it is often disregarded in theoretical models when deriving equilibrium strategies. We introduce a maximal budget that restricts the bidding actions of participants, thereby reflecting a fixed budget constraint for all valuations.

Furthermore, we investigate the impact of interdependencies among bidders' valuations. Specifically, we use the well-known *common* and *affiliated* values settings in sequential auctions (Krishna, 2009). In both settings, all bidders have the same valuation $v$ but only receive a noisy signal in the form of an observation $x_i$. In the common values setting, the valuation $v$ is drawn uniformly from the interval $[0, 1]$, while the bidders' observations, denoted as $x_i$, are uniformly drawn from $[0, 2v]$.

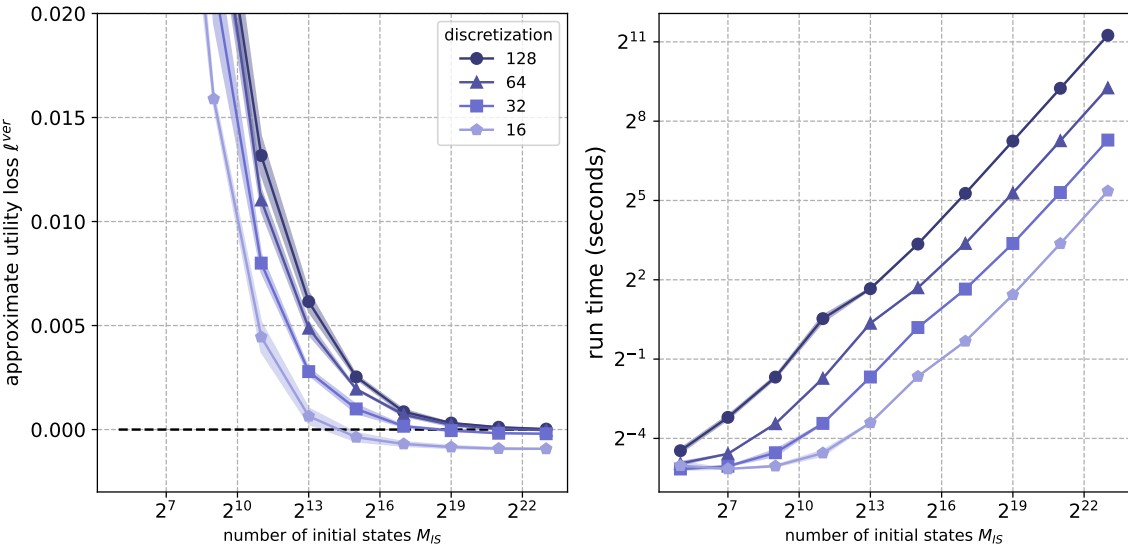

*Figure 2.* Approximate utility loss for different configurations of the discretization size and the number of simulations *(left)* and their corresponding run times *(right)*.

*Table 3.* Approximated utility losses of the experiments with valuation interdependencies and risk-averse bidders. Here, no analytical equilibrium is available for comparion.

| mechanism | prior | risk $\rho$ | metric | REINFORCE | PPO |
|---|---|---|---|---|---|
| first-price | affiliated | 0.25 | $\ell^{\text{ver}}$ | 0.0013 (0.0004) | -0.0004 (0.0001) |
| | | 0.50 | $\ell^{\text{ver}}$ | 0.0008 (0.0002) | -0.0002 (0.0001) |
| | | 0.75 | $\ell^{\text{ver}}$ | 0.0005 (0.0001) | -0.0001 (0.0001) |
| second-price | common | 0.25 | $\ell^{\text{ver}}$ | 0.0014 (0.0004) | 0.0029 (0.0029) |
| | | 0.50 | $\ell^{\text{ver}}$ | 0.0010 (0.0002) | 0.0026 (0.0020) |
| | | 0.75 | $\ell^{\text{ver}}$ | 0.0011 (0.0006) | 0.0029 (0.0023) |

On the other hand, in the standard uniform affiliated values model, the common valuation is determined by the mean of all observations $x_i$, while the individual observations are given by $x_i = z_i + s$, where $z_i$ and $s$ are uniformly drawn from $[0, 1]$, resulting in correlated observations.

It is worth noting that each of the aforementioned adaptations significantly increases the complexity of analytical derivations, often leading to isolated studies of their individual effects. However, our approach allows us to explore settings that incorporate multiple effects simultaneously, showcasing the versatility of our approach.

Table 3 shows estimated utility losses in the common and affiliated values settings, considering different levels of risk aversion. Additionally, Figure 3 presents an approximate equilibrium strategy in the common values setting with a second-price rule and a risk parameter of $\rho = 0.25$. One observes a significant increase in bids from the first to the second stage of the auction. Presumably, this can be attributed to two factors. Firstly, bidders have a second chance to win the item in the second stage, leading to lower bids in the first stage. Secondly, reaching the second stage means that the first stage's winner had a higher estimation of the true value, leading bidders to perceive their initial estimation as an underestimation of the true valuation. Consequently, they bid more aggressively in the second stage.

Table 4 presents the estimated utility losses under different levels of budget constraints in the affiliated values setting. To illustrate an approximate equilibrium strategy in the presence of budget constraints, Figure 4 showcases a scenario involving a second-price auction with a budget of $0.8$. Similar to our previous findings, we observe a pattern where bids increase from the first to the second stage of the auction. This behavior can be attributed to the same underlying reasons identified in the common values experiment. Bidders place lower initial bids in the first stage, as they have the opportunity to win in

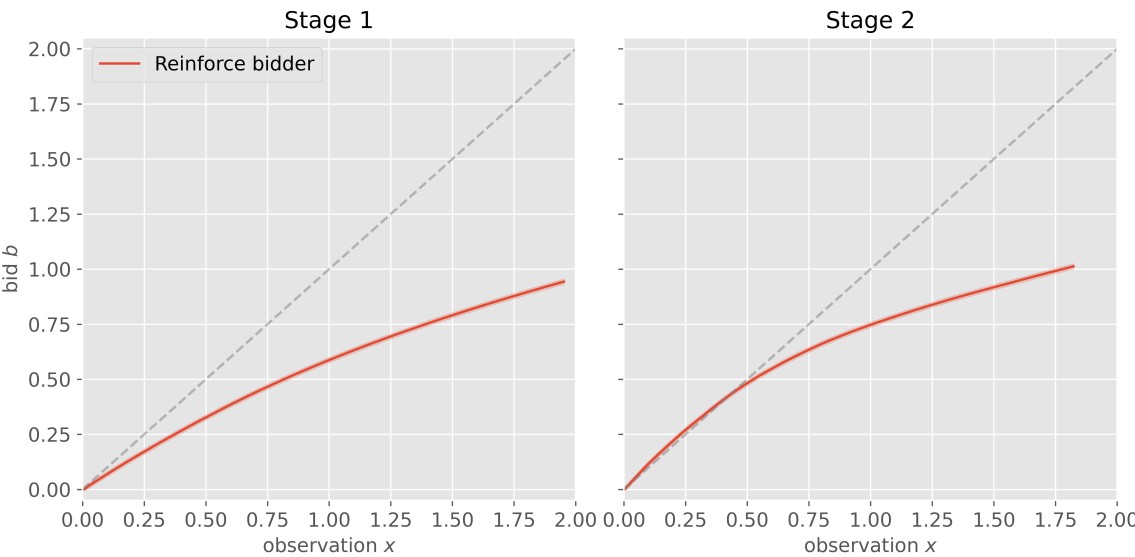

*Figure 3.* REINFORCE-based learned strategies in sequential sales with a second-price mechanism, a common values among the agents, and three risk-averse bidders with $\rho = 0.25$ in two stages.

*Table 4.* Approximated utility losses in a two-stage sequential auction with affiliated values and budget constrainted bidders.

| mechanism | budget | metric | REINFORCE | PPO |
|---|---|---|---|---|
| first-price | 0.6 | $\ell^{\text{ver}}$ | 0.0058 (0.0002) | 0.0058 (0.0002) |
| | 0.8 | $\ell^{\text{ver}}$ | 0.0104 (0.0002) | 0.0069 (0.0003) |
| second-price | 0.6 | $\ell^{\text{ver}}$ | 0.0045 (0.0002) | 0.0043 (0.0002) |
| | 0.8 | $\ell^{\text{ver}}$ | 0.0039 (0.0001) | 0.0036 (0.0002) |

the second stage if they lose initially. Additionally, the knowledge that another bidder won in the first stage with a higher estimation of the true valuation prompts bidders to increase their own estimation.

### C.4  Sequential Sales: Asymmetric Equilibria

In previous experiments, the focus was on the symmetric setting. Here, each agent faces the identical decision problem and, therefore, as is commonly assumed in the literature, all agents employ the same strategy in equilibrium (Krishna, 2009). We model this by letting the agents share a single neural network.

We relax this by allowing each agent to employ a different strategy. Specifically, we consider a two-stage second-price sequential auction with uniform prior and risk-neutral bidders, where each agent trains its own neural network. Our empirical findings show that both the REINFORCE and PPO algorithms consistently converge to an approximate asymmetric equilibrium, as depicted in Figure 5. The estimated utility loss is very low for all agents (see Table 5).

In this equilibrium, two bidders utilize almost identical strategies, which significantly deviate from the symmetric equilibrium strategy. The remaining bidder places slightly higher bids for valuations below 0.15 and then maintains almost constant bids, resulting in significantly lower bids than from the other two bidders for higher valuations. In the second stage, all agents employ a truthful strategy.

Interestingly, the estimated utility for each agent is nearly identical, with a mean of 0.2476 for the lower bidding agent and 0.2466 for the two higher bidding ones. Furthermore, the expected utility closely matches the expected utility in the symmetric equilibrium, which is 0.25 analytically and 0.2465 in the approximated equilibrium reported in Section 5. Notably, we did not observe a similar effect in sequential auctions using a first-price rule, where all bidders converge approximately to the symmetric equilibrium.

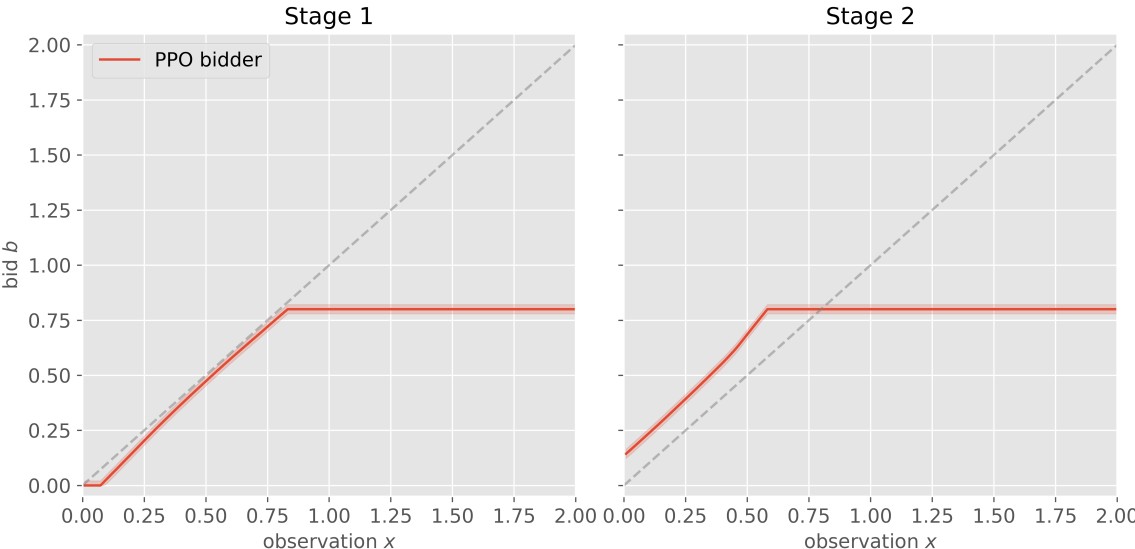

*Figure 4.* REINFORCE-based learned strategies in sequential sales with affiliated values under the second-price mechanism with a budget constraint of 0.8, two-stages, and three bidders.

*Table 5.* Approximated utility losses for all agents $i$ in a second-price two-stage auction with three bidders that end up in an asymmetric equilibrium.

| $i$ | metric | REINFORCE | PPO |
|-----|--------|-----------|-----|
| 1 | $\ell^{\text{ver}}$ | 0.0009 (0.0003) | 0.0006 (0.0003) |
| 2 | $\ell^{\text{ver}}$ | 0.0008 (0.0003) | 0.0007 (0.0001) |
| 3 | $\ell^{\text{ver}}$ | 0.0009 (0.0002) | 0.0005 (0.0002) |

This empirical observation is particularly intriguing as it shows the existence of an approximate asymmetric equilibrium, even when the overall decision problem is symmetric. Furthermore, in this case, the approximate asymmetric equilibrium is attracting under simple learning dynamics.

## C.5 Elimination Contests

Contests are used to model lobbying, political campaigns, and R&D competitions, among others (Konrad et al., 2009; Corchón et al., 2018). Equilibrium analysis has been a central approach to analyzing competition in contests. Almost the entire literature focuses on Nash equilibria and models contests as complete-information games (Corchón et al., 2018). However, similar to auction theory, much of the strategic complexity in contests is due to the fact that contestants only have incomplete information about their competitors, and they aim to find a bid function for a continuous set of signals.

Importantly, here we analyze multi-stage contests. In many real-life contests, players are initially divided into a few groups, they first compete within their subgroups, and then winners from each group compete again in later stages. The central question is whether pre-commitments in sequential or elimination contests incentivize players to invest more effort than in a single-stage contest. Furthermore, it is important to understand the effects of information revelation between stages.

We study the signaling contest as introduced by Zhang (2008) as it has some interesting properties. He analyzes the effect of publishing private information (valuations) or submitted bids (signaling effect) on the equilibrium in a two-round elimination contest. The work is inspired by Moldovanu & Sela (2006) who leave open to analyze the role of information in contests with multiple rounds. While the paper studies a fairly general case, the equilibrium strategies are provided as abstract integral solutions. Therefore, we consider a special instance where we can solve the related integrals and provide an explicit equilibrium strategy.

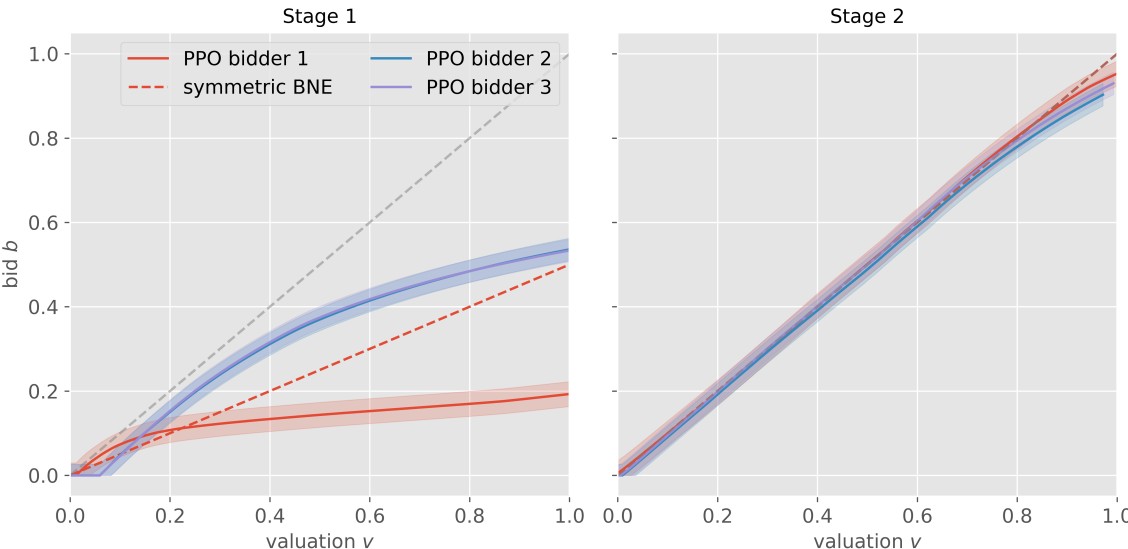

*Figure 5.* Asymmetric PPO-based learned strategies in sequential sales with a second-price mechanism, two stages, and three bidders.

Consider $N = 4$ risk-neutral bidders that privately learn their valuations $v = (v_1, v_2, v_3, v_4)$ for the prize, which are independently and uniformly distributed on the interval $[1.0, 1.5]$. In the first stage, they compete within two equally sized groups of two bidders by simultaneously submitting bids. Here, all bidders pay for their bids regardless of success (all-pay auction). Then, the two winners compete in the final round. Before the final round, either their true valuations or their bids are revealed to the others. Thus, the finalists can now base their decisions on their private information about the prize and the public information about their opponent. In the second round, the players' winning probabilities are equal to the ratio of their own bid to the cumulative bids of the finalists (Tullock contest).

To reduce sample variance, we directly model their utility as the expected utility for given valuations and efforts by weighting their valuation for the prize by the probability of winning. For a finalist $i$, we have $u_i(v_i, a_i, a_{-i}) = \frac{a_{i2}}{a_{i2} + a_{j2}} v_i - a_{i2} - a_{i1}$, where $j$ denotes the other finalist, and for a non-finalist $k$, we have $u_k(v_k, a_k, a_{-k}) = -a_{k1}$. Zhang (2008) derived the following equilibrium:

**Proposition C.2** (Zhang (2008)). *Consider a four-bidder two stage-contest as described above. Let $i$ be some bidder, and denote with $j$ the first round's winner of the other group. Then there exists a separating equilibrium for both information cases, which is given by the following.*

1. *Assuming the true valuations are revealed after the first stage, i.e., $\sigma_{i2}(a_{\cdot 1}) = v_j$, we have the following symmetric equilibrium:*

$$\beta_{i1}(v_i) = WE(v_i)$$

$$\beta_{i2}(v_i, v_j) = \frac{v_i^2 v_j}{(v_i + v_j)^2}$$

2. *Assuming the winning bids of the other group are revealed after the first stage, i.e., $\sigma_{i2}(a_{\cdot 1}) = a_{1j}$, we have the following equilibrium:*

$$\beta_{i1}(v_i) = WE(v_i) + SE(v_i)$$

$$\beta_{i2}(v_i, a_{j1}) = \frac{v_i^2 \beta_{i1}^{-1}(a_{j1})}{\left(v_i + \beta_{i1}^{-1}(a_{j1})\right)^2}$$

*where the functions WE and SE are defined as follows:*

$$WE(v_i) = 27 \log\left(v_i + \frac{3}{2}\right) - \frac{17 v_i}{2} - \frac{43 \log\left(\frac{5}{2}\right)}{4} + \frac{7 v_i^2}{2} - 2 v_i^3 - 4 \log\left(v_i + 1\right)\left(v_i^4 - 1\right)$$

*Table 6.* Learning results in the signaling contest. We again report the mean $L_2$ loss for both stages and the utility losses $\ell^{\text{equ}}$ and $\ell^{\text{ver}}$ and their standard deviations over ten runs.

| public information | metric | REINFORCE | PPO |
|---|---|---|---|
| valuations | $\ell^{\text{equ}}$ | 0.0001 (0.0002) | 0.0000 (0.0001) |
| | $\ell^{\text{ver}}$ | 0.0013 (0.0004) | -0.0003 (0.0003) |
| | $L_2^{S1}$ | 0.0059 (0.0015) | 0.0029 (0.0011) |
| | $L_2^{S2}$ | 0.0060 (0.0010) | 0.0013 (0.0004) |
| bids | $\ell^{\text{equ}}$ | 0.0002 (0.0001) | 0.0000 (0.0001) |
| | $\ell^{\text{ver}}$ | -0.0008 (0.0004) | 0.0000 (0.0004) |
| | $L_2^{S1}$ | 0.0072 (0.0012) | 0.0029 (0.0008) |
| | $L_2^{S2}$ | 0.0066 (0.0010) | 0.0014 (0.0003) |

$$+ 4 \log\left(v_i + \frac{3}{2}\right)\left(v_i^4 - \frac{81}{16}\right) + 7$$

$$SE(1) = 0$$

$$SE(v_i) = 17 \log(5) - 8 \log(v_i + 1) - 9 \log\left(v_i + \frac{3}{2}\right) - 17 \log(2) - 16\, v_i + 8\, v_i^2 \log(v_i + 1)$$

$$+ 16\, v_i^3 \log(v_i + 1) - 16\, v_i^4 \log(v_i + 1) - 8\, v_i^2 \log\left(v_i + \frac{3}{2}\right) - 16\, v_i^3 \log\left(v_i + \frac{3}{2}\right)$$

$$+ 16\, v_i^4 \log\left(v_i + \frac{3}{2}\right) - \frac{135}{2\, v_i + 3} + 18\, v_i^2 - 8\, v_i^3 + 33 \; \text{for } v_i \in (1, 1.5].$$

The results are presented in Table 6 and again show a very small utility loss. If the estimated utility loss is negative, this means the learned strategy is better than the best finite precision step function strategy.

## C.6 Stackelberg Bertrand Competition

Stackelberg games, originally introduced by Von Stackelberg in 1934, hold significant importance in economic theory and find various applications today (Li, 1985; Dowrick, 1986; Powell, 2007). These game models involves a two-step process, where a leader makes the initial move, which is then observed by a follower who subsequently decides on its action. One crucial question that arises is the relative advantage of being the leader or the follower in such scenarios.

In our study, we focus on a specific variation known as the Stackelberg duopoly with incomplete information, as introduced by Arozamena & Weinschelbaum (2009). Their work explores equilibrium strategies in both simultaneous and Stackelberg-Bertrand competitions, comparing the behavior of the leader firm in these settings under different assumptions. Notably, they establish the existence of a second-mover advantage in the Stackelberg setting. While their paper provides a general analysis and implicitly presents the equilibrium solution by providing its inverse, we delve into a special case within this framework.

Consider a Stackelberg-Bertrand competition with two firms competing in a homogeneous-good market. Each firm sets its price, and the goal is to investigate the strategic interactions between the leader (Firm 1) and the follower (Firm 2). Let $c_1$ and $c_2$ represent the unit costs of firms 1 and 2, respectively, which are drawn independently and identically distributed from the cumulative distribution function $F(c) = \frac{1}{2}c + \frac{c^2}{2}$. These costs are considered private information for each firm.

The game unfolds in the following manner: Firm 1 observes its private cost $c_1$ and subsequently sets its price $p_1$. Firm 2 observes its private cost $c_2$ and also the leader's posted price $p_1$. Based on this information, Firm 2 then sets its own price $p_2$. Firm 1 wins the competition if $p_1 < p_2$, otherwise Firm 2 wins. The loser gets a utility of zero, whereas the winner $i$ receives a utility of $u_i(c_i, p_1, p_2) = (p_i - c_i) \cdot Q(p_i)$, where $Q(p) = 10 - p$ denotes the demand and $p_i = \min\{p_1, p_2\}$. Arozamena & Weinschelbaum (2009) derived the following class of equilibria.

**Proposition C.3** (Arozamena & Weinschelbaum (2009)). *Consider a two-firm Stackelberg-Bertrand competition as described above. Then for every measurable function $f : \mathbb{R} \to \mathbb{R}$ such that $f(x) > x$, an equilibrium is given as follows:*

$$\beta_{11}^{-1}(p_1) = p_1 - \frac{Q(p_1)(1 - F(p_1))}{Q(p_1)F'(p_1) - Q'(p_1)(1 - F(p_1))} = \frac{4p_1^3 - 27p_1^2 - 24p_1 + 20}{3p_1^2 - 18p_1 - 12},$$

*Table 7.* Learning results in the Bertrand competition. We again report the $L_2$ loss for each stage and agent, and the utility losses $\ell^{\text{equ}}$ and $\ell^{\text{ver}}$ with their standard deviations over ten runs.

| agent | metric | REINFORCE | PPO |
|---|---|---|---|
| leader | $\ell^{\text{equ}}$ | 0.0006 (0.0008) | 0.0001 (0.0006) |
| | $\ell^{\text{ver}}$ | 0.0004 (0.0002) | 0.0008 (0.0003) |
| | $L_2^{S1}$ | 0.0064 (0.0036) | 0.0053 (0.0025) |
| follower | $\ell^{\text{equ}}$ | 0.0337 (0.0031) | 0.0435 (0.0078) |
| | $\ell^{\text{ver}}$ | -0.0129 (0.0032) | -0.0031 (0.0081) |
| | $L_2^{S2}$ | 0.0046 (0.0005) | 0.0059 (0.0010) |

$$\beta_{22}(c_2, p_1) = \begin{cases} \min\{p_1, p^M(c_2)\}, & \text{for } p_1 \geq c_2, \\ f(b_1), & \text{else,} \end{cases}$$

*where $p^M(c_2) = \max_{p_2} Q(p_2)(p_2 - c_2) = 5 + \frac{c_2}{2}$ denotes the monopoly price, and $\beta_{11}^{-1}$ is the leader's inverse equilibrium strategy.*

The leader's equilibrium strategy $\beta_{11}$ is guaranteed to be invertible in the above setting, so that we recover it by numerically inverting $\beta_{11}^{-1}$ from above. The results of the algorithms under consideration in this study are presented in Table 7. They show a small $L_2$-loss and (estimated) utility loss. If the estimated utility loss is negative, this means the learned strategy is better than the best finite precision step function strategy. Note that the utility losses' specific values $\ell^{\text{equ}}$ and $\ell^{\text{ver}}$ are not directly comparable to other experiments as the equilibrium utilities are about ten to fifteen times larger in this experiment.

# D  Verification

Let us outline our methodology for verifying approximate equilibria in continuous multi-stage games. First, we describe some preliminaries needed for the main theorem on its appropriateness. Next, we introduce the verification procedure formally and provide a proof for the main theorem.

## D.1  Preliminaries

In Theorem A.1, we defined a multi-stage game with continuous signals and actions. In what follows, we summarize some additional assumptions required for bounding the error of our verifier.

Throughout, we assume the game to have *perfect recall*. This means that players remember all information they received, and in particular, their own actions. This assumption greatly simplifies theory and is usually made in literature (Myerson & Reny, 2020).

**Definition D.1** (Perfect recall). Let $\Gamma = (\mathcal{N}, T, S, \mathcal{A}, p, \sigma, u)$ be a multi-stage game. It is said to have *perfect recall* if for all $it \in L$ and $r > t$, there are measurable functions $\Psi : S_{ir} \to S_{it}$ and $\psi : S_{ir} \to A_{it}$ such that $\Psi(\sigma_{ir}(a_{<r})) = \sigma_{it}(a_{<t})$ and $\psi(\sigma_{ir}(a_{<r})) = a_{it}$, for all $a \in \mathcal{A}$.

Under perfect recall, one can extract all of $i$'s actions and received signals up to that point from a signal $s_{ir} \in S_{ir}$. That is, there exist functions $\psi_{irt} : S_{ir} \to \mathcal{A}_{it}$ and $\Psi_{irt} : S_{ir} \to S_{it}$ such that $\psi_{irt}(\sigma_{ir}(a_{<r})) = a_{it}$ and $\Psi_{irt}(\sigma_{ir}(a_{<r})) = \sigma_{it}(a_{<t})$ for all $a_{<r} = (a_{\cdot 1}, \cdots, a_{\cdot r-1}) \in \mathcal{A}_{<r}$. Denote with $\psi_{ir} = (\psi_{ir1}, \psi_{ir2}, \ldots, \psi_{irr-1})$ and $\Psi_{ir} = (\Psi_{ir1}, \Psi_{ir2}, \ldots, \Psi_{irr-1})$ the corresponding mappings from $S_{ir}$ into $\mathcal{A}_{i<r}$ and from $S_{ir}$ into $S_{i<r}$ respectively.

Later on, we are interested in two restrictions of the strategy spaces and their intersection. The first restriction allows only pure strategies, i.e., strategies that map onto Dirac measures for a given signal, which we denote by

$$\Sigma_{it}^{\text{p}} := \{\beta \in \Sigma_{it} \mid \beta(s_{it}) = \delta_a \text{ for } s_{it} \in S_{it} \text{ and } a \in \mathcal{A}_{it}\}. \tag{6}$$

We can identify this with the set of measurable functions from signals to actions. With slight abuse of notation, we denote both sets by $\Sigma_{it}^{\text{p}}$ and note where it does not become clear from the context. We set $\Sigma_i^{\text{p}} = \bigtimes_{t \in T} \Sigma_{it}^{\text{p}}$.

For verification, we are interested in whether one can achieve the same best-response utility by restricting the space of a single-agent to pure strategies. This is often satisfied under mild assumptions. For example, it is satisfied in most Bayesian

games (Milgrom & Weber, 1985; Hosoya & Yu, 2022). Furthermore, it can often be guaranteed to be viable whenever a pure strategy equilibrium exists (Horst, 2005; Reny, 2011).

Another important restriction is the set of Lipschitz continuous functions, which we denote by

$$\Sigma_{it}^{\text{Lip}} = \{\beta_{it} \in \Sigma_{it} \mid \beta_{it} : S_{it} \to \Delta(\mathcal{A}_{it}) \text{ is Lipschitz continuous in } d_W\}, \tag{7}$$

where $d_W$ denotes the Wasserstein distance (see Theorem E.5). We set $\Sigma_i^{\text{Lip}} = \bigtimes_{t \in T} \Sigma_{it}^{\text{Lip}}$. Note that common function approximators for distributional strategies, such as neural networks, fall into the space of Lipschitz continuous strategies.

Finally, we consider the intersection of pure and Lipschitz continuous strategies, which we denote by $\Sigma_{it}^{\text{Lip, p}} := \Sigma_{it}^{\text{Lip}} \cap \Sigma_{it}^{\text{p}}$, and $\Sigma_i^{\text{Lip, p}} := \bigtimes_{t \in T} \Sigma_{it}^{\text{Lip, p}}$.

For a given $\beta_{.t}$, one can define a probability distribution from stage $t$ to $t + 1$. Let $B \subset \mathcal{A}_{.t}$ be measurable and $a_{<t} \in \mathcal{A}_{<t}$, then the mapping $P_t$ defines a transition probability from $\mathcal{A}_{<t}$ into the set of measurable subsets of $\mathcal{A}_{.t}$ by

$$P_t(B \mid a_{<t}, \beta_{.t}) = p_t(B_{0t} \mid a_{<t}) \prod_{i \in \mathcal{N}} \beta_{it}(B_{it} \mid \sigma_{it}(a_{<t})), \tag{8}$$

where $\beta_{it}(B_{it} \mid \sigma_{it}(a_{<t}))$ is the probability that player $i$ takes actions from $B_{it}$ when receiving the signal $\sigma_{it}(a_{<t})$. The players need to reason about several stages. Therefore, we inductively define probabilities that describe events from the beginning up to a certain stage.

Let $P_{<1}(\{\emptyset\} \mid \beta) = 1$, and for all $t \in T$ and measurable $B \subset \mathcal{A}_{<t+1}$, define the rollout measure up to stage $t$ under strategy profile $\beta$ as

$$P_{<t+1}(B \mid \beta) = \int_{\mathcal{A}_{<t}} P_t(\{a_{.t} : (a_{<t}, a_{.t}) \in B\} \mid a_{<t}, \beta_{.t}) \, dP_{<t}(a_{<t} \mid \beta). \tag{9}$$

Intuitively speaking, this defines the probability of an intermediate history $B$ up to stage $t$ to occur when all players act according to $\beta$. The probability measure $P(\cdot \mid \beta) := P_{<T+1}(\cdot \mid \beta)$ denotes the probability measure over outcomes induced by the strategy profile $\beta$.

Finally, player $i$'s ex-ante utility is defined as the expected utility over all possible outcomes of the game and is given by

$$\tilde{u}_i(\beta) = \int_{\mathcal{A}} u_i(a) dP(a \mid \beta). \tag{10}$$

In the game's interim stages, the individual agent reasons about what action to take after receiving a signal. To describe this optimization problem, we introduce the conditional probabilities and utilities to describe the expected utility given a certain signal and strategies. Let $\beta \in \Sigma$ be a strategy profile, $it \in L$, and $Z \subset S_{it}$ measurable, then define

$$P_{it}(Z \mid \beta) = P_{<t}(\sigma_{it}^{-1}(Z) \mid b) = P_{<t}(\{a_{<t} : \sigma_{it}(a_{<t}) \in Z\} \mid \beta), \tag{11}$$

to be the probability that player $i$'s date $t$ signal is in $Z$ under strategy profile $\beta$. Consequently, for any $it \in L$ and measurable $Z \subset S_{it}$ such that $P_{it}(Z \mid \beta) > 0$, we define, with a slight abuse of notation, the conditional probabilities as

$$P_{<t}(B \mid Z, \beta) = P_{<t}(B \cap \sigma_{it}^{-1}(Z) \mid \beta) / P_{it}(Z \mid \beta) \quad \forall B \subset \mathcal{A}_{<t} \text{ measurable}, \tag{12}$$

and

$$P(B \mid Z, \beta) = P(\{a \in B : \sigma_{it}(a_{<t}) \in Z\} \mid \beta) / P_{it}(Z \mid \beta) \quad \forall B \subset \mathcal{A} \text{ measurable}. \tag{13}$$

With all of this, the conditional expected utilities for a measurable $Z \subset S_{it}$ are defined as

$$\tilde{u}_i(\beta \mid Z) = \int_{\mathcal{A}} u_i(a) dP(a \mid Z, \beta). \tag{14}$$

### D.2 Verification Procedure

The verification procedure of an agent's learned strategy consists of two main parts. First, the strategy space must be discretized such that the search space is reduced to finite size, and the number of simulations must be set such that the expected utilities (across nature and the opponents' action probabilities) can be approximated via sampling. Second, the deployment of all possible strategies is simulated, and the best-performing strategy with the highest utility is compared to the utility of the actual strategy.

#### D.2.1 FINITE PRECISION STEP-FUNCTIONS

In our approach, we aim to restrict the search for a best-response from the view of a single agent to a finite set. To achieve this, it is necessary to ensure that a finite discretization adequately captures both the signal and action spaces. To facilitate our theoretical analysis and ensure the practicality of our approach, we make the following assumption.

**Assumption D.2.** For every $it \in L^*$, we assume there exist finitely many bounded closed intervals $\mathcal{A}_{it}^r$ and $S_{it}^r$, such that $\mathcal{A}_{it} = \bigtimes_{r=1}^{N_{\mathcal{A}_{it}}} \mathcal{A}_{it}^r$ and $S_{it} = \bigtimes_{r=1}^{N_{S_{it}}} S_{it}^r$ for dimensions $N_{\mathcal{A}_{it}}, N_{S_{it}} \in \mathbb{N}$.

Under this assumption, we can divide the signal *and* action spaces into grids. For $it \in L$, we denote the grid points of the signal and action spaces as $S_{it}^{\mathrm{D}}$ and $\mathcal{A}_{it}^{\mathrm{D}}$, respectively. Here, $\mathrm{D} \in \mathbb{N}$ denotes a precision parameter, where an increasing D translates to an increase in grid points. Furthermore, for every $it \in L$ and $\mathrm{D} \in \mathbb{N}$ there exists a finite number of disjoint grid cells that consist of a product of half-open intervals, partitioning $S_{it}$. We denote these grid cells by $C_{it}^k$, where $1 \le k \le G_{S_{it}}^{\mathrm{D}}$ and $G_{S_{it}}^{\mathrm{D}} \in \mathbb{N}$ is the number of grid cells. Finally, let the tuple $\mathcal{G}_{it} = \left( S_{it}^{\mathrm{D}}, \mathcal{A}_{it}^{\mathrm{D}}, C_{it}, \mathrm{D} \right)$ denote the signal and action space discretization for $it \in L$.

We define the set of step functions with precision D for $it \in L$ by

$$\Sigma_{it}^{\mathrm{D}} \left( \mathcal{G}_{it} \right) := \left\{ s_{it} \mapsto \sum_{k=1}^{G_{S_{it}}^{\mathrm{D}}} \chi_{C_{it}^k}(s_{it}) a_{it}^k \,\middle|\, a_{it}^k \in \mathcal{A}_{it}^{\mathrm{D}} \right\}, \tag{15}$$

and $\Sigma_i^{\mathrm{D}} \left( \mathcal{G}_i \right) = \bigtimes_{t \in T} \Sigma_{it}^{\mathrm{D}} \left( \mathcal{G}_{it} \right)$. Any grid so that $\Sigma_{it}^{\mathrm{D}}$ can approximate any pure Lipschitz continuous function well for sufficiently high D (see Theorem E.4) can be used. In the following, we restrict ourselves to the regular grid and show that it satisfies this property. Therefore, we drop the discretization and write $\Sigma_i^{\mathrm{D}}$ instead of $\Sigma_i^{\mathrm{D}} \left( \mathcal{G}_{it} \right)$.

For any given finite precision $\mathrm{D} \in \mathbb{N}$, we make a *discretization error*, which we denote by

$$\varepsilon_{\mathrm{D}} := \sup_{\beta_i' \in \Sigma_i^{\mathrm{Lip,\,p}}} \tilde{u}_i(\beta_i', \beta_{-i}) - \sup_{\beta_i' \in \Sigma_i^{\mathrm{D}}} \tilde{u}_i(\beta_i', \beta_{-i}). \tag{16}$$

#### D.2.2 BACKWARD INDUCTION OVER FINITE PRECISION STEP FUNCTIONS

For every finite precision D, there are finitely many elements in $\Sigma_i^{\mathrm{D}}$, which can be translated into finitely many decision points for player $i$. That is, we can build a finite game or decision tree representing all possible step functions from $\Sigma_i^{\mathrm{D}}$. We perform a backward induction scheme on this finite decision tree to get the maximal ex-ante utility from any step function.

To achieve this, we define player $i$'s *counterfactual conditional utility* as the conditional utility for taking a specific action given a certain signal, excluding player $i$'s influence of reaching this signal. This is similar to formulations of counterfactual reach probabilities and utilities from literature in finite games (Zinkevich et al., 2007). Before the counterfactual conditional utilities can be formally defined, we need to introduce some other objects first.

We exclude player $i$'s influence of reaching a certain signal grid cell $C_{it}^k \subset S_{it}$ by considering a strategy that deterministically plays to reach $C_{it}^k$. That is possible because, without loss of generality, one can assume that there exists a unique sequence of grid actions $a_{i<t}^{C_{it}^k} \in \mathcal{A}_{i<t}^{\mathrm{D}}$ that need to be taken for every grid cell $C_{it}^k$. To see this, note that due to perfect recall, agent $i$'s actions taken prior to stage $t$ can be extracted from every signal $s_{it}$. Due to our construction, this is a unique sequence for every grid cell $C_{it}^k$ if, for example, each action space $\mathcal{A}_{ir}$ gets appended to the signaling space in the following stage $S_{ir+1}$. Therefore, for every $C_{it}^k$, there exist $s_{it} \in C_{it}^k$ and $a_{i<t} \in \mathcal{A}_{i<t}^{\mathrm{D}}$ such that $\psi_{it}(s_{it}) = a_{i<t}$. At the same time, there exists no $s_{it}' \in C_{it}^k$ such that $\psi_{it}(s_{it}') \ne a_{i<t}$ and $\psi_{it}(s_{it}') \in \mathcal{A}_{i<t}^{\mathrm{D}}$. Therefore, we define functions $\psi_{it}^{\mathrm{D}}(C_{it}^k) = a_{i<t}^{C_{it}^k}$ that map a grid cell to its unique history of grid actions.

Given a finite precision step function strategy $\beta_i \in \Sigma_i^{\mathrm{D}}$, we can now construct a strategy for player $i$ that plays to reach $C_{it}^k$ (adapting stages $1, \ldots, t-1$), takes a certain action in stage $t$, and remains the same for stages $t+1, \ldots, T$. More specifically, let $it \in L$, $\beta = (\beta_i, \beta_{-i})$ be a strategy profile with $\beta_i \in \Sigma_i^{\mathrm{D}}$ and $\beta_{-i} \in \Sigma_{-i}$, and $a_{it}^k \in \mathcal{A}_{it}^{\mathrm{D}}$. Then we define a function $(\beta_i)^{C_{it}^k, a_{it}^k} = \left( (\beta_{i1})^{C_{it}^k, a_{it}^k}, \ldots, (\beta_{iT})^{C_{it}^k, a_{it}^k} \right)$ that is playing to reach $C_{it}^k$ in the following way:

$$(\beta_{ir})^{C_{it}^k, a_{it}^k} = \beta_{ir} \quad \text{for } r > t,$$

$$(\beta_{it})^{C_{it}^k, a_{it}^k}(s_{it}) = \begin{cases} a_{it}^k & \text{for } s_{it} \in C_{it}^k, \\ \beta_{it}(s_{it}) & \text{for } s_{it} \in S_{it} \setminus C_{it}^k, \end{cases}$$

$$(\beta_{i<t})^{C_{it}^k, a_{it}^k}(\Psi_{it}(s_{it})) = \begin{cases} \psi_{it}^{\mathrm{D}}(C_{it}^k) & \text{for } s_{it} \in C_{it}^k, \\ \beta_{i<t}(\Psi_{it}(s_{it})) & \text{for } s_{it} \in S_{it} \setminus C_{it}^k, \end{cases}$$

The counterfactual conditional utilities for precision D are then defined as

$$\tilde{u}_i^{\mathrm{c,\,D}}\left( \beta \mid C_{it}^k, a_{it}^k \right) = \tilde{u}_i\left( (\beta_i)^{C_{it}^k, a_{it}^k}, \beta_{-i} \mid C_{it}^k \right), \tag{17}$$

where $\tilde{u}_i(\,\cdot \mid \cdot\,)$ is the conditional utility defined in Equation 14. Note that $\tilde{u}_i^{\mathrm{c,\,D}}$ is independent of $\beta_{i<t} \in \Sigma_{i<t}^{\mathrm{D}}$, as only histories conditioned on observing signals from $C_{it}^k$ are considered. All actions that may be taken off paths that lead to this set of signals do not matter. Therefore, we write $\tilde{u}_i^{\mathrm{c,\,D}}\left( \beta_{i>t}, \beta_{-i} \mid C_{it}^k, a_{it}^k \right)$ instead of $\tilde{u}_i^{\mathrm{c,\,D}}\left( \beta \mid C_{it}^k, a_{it}^k \right)$, where $\beta_{i>t} = (\beta_{it+1}, \ldots, \beta_{iT})$, to emphasize this independence. We are now ready to define a best response over the finite strategy set $\Sigma_i^{\mathrm{D}}$ via backward induction. For a given opponent strategy profile $\beta_{-i}$, we inductively define a step function $\beta_i^{\mathrm{D},*} \in \Sigma_i^{\mathrm{D}}$. For the last stage $s_{iT} \in S_{iT}$, define

$$\beta_{iT}^{\mathrm{D},*}(s_{iT}) = \underset{a_{iT} \in \mathcal{A}_{iT}^{\mathrm{D}}}{\arg\max}\ \tilde{u}_i^{\mathrm{c,\,D}}\left( \beta_{-i} \mid C_{iT}^k, a_{iT} \right), \tag{18}$$

where $C_{iT}^k$ is the unique set such that $s_{iT} \in C_{iT}^k$. For preceding stages $t < T$, we define

$$\beta_{it}^{\mathrm{D},*}(s_{it}) = \underset{a_{it} \in \mathcal{A}_{it}^{\mathrm{D}}}{\arg\max}\ \tilde{u}_i^{\mathrm{c,\,D}}\left( \beta_{i>t}^{\mathrm{D},*}, \beta_{-i} \mid C_{it}^k, a_{it} \right), \tag{19}$$

again with $s_{it} \in C_{it}^k$. Note that the $\arg\max_{a_{it} \in \mathcal{A}_{it}^{\mathrm{D}}}$ is non-empty, as the utility functions are bounded, and there are only finitely many values to consider. If there is more than one element in the $\arg\max_{a_{it} \in \mathcal{A}_{it}^{\mathrm{D}}}$, then we simply choose one discrete action for a whole grid cell $C_{it}^k$. This backward induction procedure gives us a best response over the set $\Sigma_i^{\mathrm{D}}$ (see Theorem E.2).

### D.2.3 MONTE-CARLO INTEGRATION FOR CONDITIONAL UTILITIES

The backward induction procedure above assumes that the conditional expected utilities from Equations 18 and 19 can be evaluated, which is, in general, impossible. It would require having access to the expectations of the conditional utilities (Equation 14) for which there are no closed-form solutions in general. Therefore, we employ Monte-Carlo approximation to estimate $\beta_i^{\mathrm{D},*}$ and its ex-ante utility. We separate the approximation into a simulation and an aggregation phase.

We start with the simulation phase by sampling a single initial game state, and the players receive their respective signals $s_{\cdot 1}$. We collect the opponent actions $a_{-i1}$ according to $\beta_{-i1}$. For player $i$, we register into which grid cell $C_{i1}^k$ the signal $s_{i1}$ belongs and increase the cell's counter (aka. visitation count), which we denote by $M(C_{i1}^k)$. Then, we simulate the transition to the next stage for every possible action $a_{i1} \in \mathcal{A}_{i1}^{\mathrm{D}}$, multiplying the number of simulated games by a factor of $|\mathcal{A}_{i1}^{\mathrm{D}}|$. We proceed in this pattern; for each simulation, collect the opponent actions according to $\beta_{-it}$, register the corresponding grid cell $C_{it}^k$ for player $i$'s signal $s_{it}$, increase a respective counter $M(C_{it}^k)$, and simulate the state transition for every possible action $a_{it} \in \mathcal{A}_{it}^{\mathrm{D}}$ multiplying the number of simulated games by a factor of $|\mathcal{A}_{it}^{\mathrm{D}}|$.[1] After $T$ stages, there are $\prod_{t \in T} |\mathcal{A}_{it}^{\mathrm{D}}|$ complete histories $a$, for which the utility $u_i(a)$ is evaluated. This procedure is performed for $M_{\mathrm{IS}} \in \mathbb{N}$ initial states,

---

[1] One can decrease this branching factor sometimes using game-specific knowledge. For example, if an agent loses in the first stage of the signaling contest, he or she may no longer bid in the second stage.

resulting in a total of $M_{\text{Tot}} = M_{\text{IS}} \cdot \prod_{t \in T} |\mathcal{A}_{it}^{\text{D}}|$ simulated histories and evaluated utilities, concluding the simulation phase. We denote the set of all simulated histories by $A_{M_{\text{IS}}}$.

After performing $M_{\text{Tot}}$ simulations, the aggregation phase starts. Depending on which subsets of $A_{M_{\text{IS}}}$ are chosen, we get samples from different distributions. For example, let $\beta_i \in \Sigma_i^{\text{D}}$ arbitrary. Consider the rollout procedure above with a single initial state. Then, as we explore every discrete action exists a simulated history $a^l$ that is consistent with $\beta_i$. That is, $a_{it}^l = \beta_i(\sigma_{it}(a_{<t}^l))$ for $1 \le t \le T$. Due to construction, we have that $a^l \sim P(\cdot \mid \beta_i, \beta_{-i})$. Therefore, for every initial state, we get at least one sample for every possible $\beta_i \in \Sigma_i^{\text{D}}$.

To perform the backward induction procedure as described above, we want to sample from conditional measures as well. So, for a grid cell $C_{it}^k \subset S_{it}$ and discretized action $a_{it}^k \in \mathcal{A}_{it}^{\text{D}}$, let $\beta_i \in \Sigma_i^{\text{D}}$ such that $\beta_i = (\beta_i)^{C_{it}^k, a_{it}^k}$. That is, $\beta_i$ is playing to reach $C_{it}^k$ and then plays $a_{it}^k$. The above procedure allows us to sample $a^l \sim P(\cdot \mid \beta_i, \beta_{-i})$. Suppose $P_{it}(C_{it}^k \mid \beta_i, \beta_{-i}) > 0$ and we only consider those $\tilde{a}^l$ with $\sigma_{it}(\tilde{a}_{<t}^l) \in C_{it}^k$, then $\tilde{a}^l \sim P(\cdot \mid C_{it}^k, (\beta_i, \beta_{-i}))$. The set of simulated histories $\tilde{a}^l$ for grid cell $C_{it}^k$, discrete action $a_{it} \in \mathcal{A}_{it}^{\text{D}}$, and step functions $\beta_{i>t}^{\text{D}} \in \Sigma_{>t}^{\text{D}}$ is given by

$$A\left(\beta_{i>t}^{\text{D}}, C_{it}^k, a_{it}; M_{\text{IS}}\right) := \left\{a^l \in A_{M_{\text{IS}}} \mid \sigma_{it}(a_{<t}^l) \in C_{it}^k, a_{it}^l = a_{it}, \right.$$
$$\left. \beta_{it+m}^{\text{D}}\left(\sigma_{it+m-1}\left(a_{<t+m}^l\right)\right) = a_{it+m}^l \text{ for } 1 \le m \le T - t \right\}.$$

It holds that $\left|A\left(\beta_{i>t}^{\text{D}}, C_{it}^k, a_{it}; M_{\text{IS}}\right)\right| = M(C_{it}^k)$ for any $a_{it} \in \mathcal{A}_{it}^{\text{D}}$ and $\beta_{i>t}^{\text{D}} \in \Sigma_{>t}^{\text{D}}$. That is, we get a valid sample from $P(\cdot \mid C_{it}^k, ((\beta_{i>t})^{C_{it}^k, a_{it}^k}, \beta_{-i}))$ whenever a simulation falls into $C_{it}^k$ for every discrete action $a_{it}^k$. We define the estimated counterfactual conditional utility by

$$\hat{u}_i^{\text{c, D}}\left(\beta_{i>t}^{\text{D}}, \beta_{-i} \mid A_t\left(\beta_{i>t}^{\text{D}}, C_{it}^k, a_{it}; M_{\text{IS}}\right)\right) := \frac{1}{M(C_{it}^k)} \sum_{l=1}^{M(C_{it}^k)} u_i\left(a^l\right), \tag{20}$$

for $\beta_{i>t}^{\text{D}} \in \Sigma_{i>t}^{\text{D}}$, $a_{it} \in \mathcal{A}_{it}^{\text{D}}$, grid cell $C_{it}^k$, and $a^l \in A\left(\beta_{i>t}^{\text{D}}, C_{it}^k, a_{it}; M_{\text{IS}}\right)$. If $M(C_{it}^k) = 0$, we set the value to zero.

This approximates the counterfactual conditional utility from Equation 18. We use these to construct a step function $\beta_i^{\text{D}, M_{\text{IS}}} \in \Sigma_i^{\text{D}}$ according to the backward induction procedure from Equations 18 and 19. From this, using the relation between the counterfactual conditional and ex-ante utilities (see Lemma E.1), we get an estimated best response utility over the simulations $A_{M_{\text{IS}}}$ which we define by

$$\hat{u}_i^{\text{ver, D}}(\beta_{-i} \mid A_{M_{\text{IS}}}) := \sum_{k=1}^{G_{S_{i1}}^{\text{D}}} \frac{M(C_{i1}^k)}{\sum_j M(C_{i1}^j)} \hat{u}_i^{\text{c, D}}\left(\beta_{i>1}^{\text{D}, M_{\text{IS}}}, \beta_{-i} \mid A_1\left(\beta_{i>1}^{\text{D}, M_{\text{IS}}}, C_{i1}^k, \beta_{i1}^{\text{D}, M_{\text{IS}}}(C_{i1}^k); M_{\text{IS}}\right)\right). \tag{21}$$

In the limit, the approximation recovers the maximum utility and best response over the set of step function $\Sigma_i^{\text{D}}$ (see Lemma E.3). We denote the *simulation error* by

$$\varepsilon_{M_{\text{IS}}} := \sup_{\beta_i' \in \Sigma_i^{\text{D}}} \tilde{u}_i(\beta_i', \beta_{-i}) - \hat{u}_i^{\text{ver, D}}(\beta_{-i} \mid A_{M_{\text{IS}}}). \tag{22}$$

Finally, let $\beta \in \Sigma$ be a strategy profile. Then, we simulate $M_{\text{IS}}$ complete histories from $P(\cdot \mid \beta)$, and collect them into a data set $B_{M_{\text{IS}}}$. Using Monte-Carlo estimation, we obtain an estimation of the expected utility, which we denote by $\hat{u}_i(\beta \mid B_{M_{\text{IS}}})$. The final estimation of our verification procedure for the utility loss over pure Lipschitz continuous strategies is then given by

$$\ell^{\text{ver}}(\beta) := \hat{u}_i^{\text{ver, D}}(\beta_{-i} \mid A_{M_{\text{IS}}}) - \hat{u}_i(\beta \mid B_{M_{\text{IS}}}). \tag{23}$$

It is important to acknowledge that the computational demand of the procedure is high, primarily due to the exponential growth of the tree as the number of stages and dimensions of signal and action spaces increase. Consequently, the applicability of the procedure is limited to games with a few stages only. Nevertheless, through parallelization and various techniques that optimize the utilization of precomputed results (Johanson et al., 2011), we can achieve a high precision using a single GPU for games. For example, the runtime for two-stage games is about two minutes. A similar game with four stages takes about five hours. Note that this level of precision is sufficient for studying a wide range of relevant continuous multi-stage games. Examples of such games include Sequential Auctions (Krishna, 2009), multi-stage contests (Yildirim, 2005), or sequential Colonel Blotto Games (Powell, 2007). These games are of significant interest and can be effectively analyzed within the computational capabilities of our procedure.

### D.3 Verifier Convergence Theorem

We now introduce central assumptions and the main theorem proving convergence of our verifier. We draw on a number of auxiliary results and lemmata in the supplement E.

The second assumption required to control errors in our verifier consists of regularity assumptions on the game and the players' strategies. One assumption is that players do not respond significantly differently to slightly different signals. Additionally, we assume that the ex-post utility functions, denoted as $u_i$, are continuous. To be more precise, we make the following assumption.

**Assumption D.3.** The signaling functions $\sigma_{it} : \mathcal{A}_{<t} \to S_{it}$ are Lipschitz continuous for all $it \in L$. Furthermore, there exists $K_{0t} > 0$ such that nature's probability distribution $p_{0t}$ is $K_{0t}$-Lipschitz continuous with respect to the Wasserstein distance for every $t \in T$. More specifically, $d_W\left(p_{0t}\left(\cdot \mid a_{<t}\right), p_{0t}\left(\cdot \mid a'_{<t}\right)\right) \le K_{0t} \|a_{<t} - a'_{<t}\|$ for all $a_{<t}, a'_{<t} \in \mathcal{A}_{<t}$ and some norm $\|\cdot\|$.

**Theorem D.4.** *Let $\Gamma = (\mathcal{N}, T, S, \mathcal{A}, p, \sigma, u)$ be a multi-stage game, where Assumptions D.2 and D.3 hold, and that the utility function $u_i$ is continuous. Further, let $\beta_{-i} \in \Sigma^{Lip}_{-i}$, $\beta_i \in \Sigma_i$, and $A_{M_{IS}}$ and $B_{M_{IS}}$ be simulated data sets with initial simulation size $M_{IS} \in \mathbb{N}$ as described in Section D.2.3. Then we have that*

$$\lim_{D \to \infty} \varepsilon_D \le 0 \text{ and } \lim_{M_{IS} \to \infty} \varepsilon_{M_{IS}} = 0 \text{ almost surely.}$$

*Furthermore, we receive an upper bound on the utility loss over pure Lipschitz continuous strategies for the strategy profile $\beta = (\beta_i, \beta_{-i})$ by*

$$\lim_{D \to \infty} \lim_{M_{IS} \to \infty} \ell_i^{ver}(\beta) = \lim_{D \to \infty} \lim_{M_{IS} \to \infty} \hat{u}_i^{ver,\, D}(\beta_{-i} \mid A_{M_{IS}}) - \hat{u}_i(\beta \mid B_{M_{IS}})$$
$$\ge \sup_{\beta'_i \in \Sigma_i^{Lip,\, p}} \tilde{u}_i(\beta'_i, \beta_{-i}) - \tilde{u}_i(\beta) = \tilde{\ell}_i^{Lip,\, p}(\beta) \text{ a. s.}$$

*Proof.* By Theorem E.3, we have almost sure convergence of $\lim_{M_{IS} \to \infty} \varepsilon_{M_{IS}} = 0$. To finish the first statement, it remains to show $\lim_{D \to \infty} \varepsilon_D \le 0$.

Let $\epsilon > 0$ and $\bar{\beta}_i \in \Sigma_i^{Lip,\, p}$ such that $\sup_{\beta'_i \in \Sigma_i^{Lip,\, p}} \tilde{u}_i(\beta'_i, \beta_{-i}) - \tilde{u}_i(\bar{\beta}_i, \beta_{-i}) \le \epsilon$. Then, by Theorem E.4, there exists a sequence $\{\beta_i^D\}_{D \in \mathbb{N}}$ with $\beta_i^D \in \Sigma_i^D$ such that

$$\lim_{D \to \infty} \|\bar{\beta}_i - \beta_i^D\|_\infty = 0.$$

By Theorem E.12, we further get

$$\lim_{D \to \infty} d_W\left(P\left(\cdot \mid \bar{\beta}_i, \beta_{-i}\right), P\left(\cdot \mid \beta_i^D, \beta_{-i}\right)\right) = 0.$$

The utility functions $u_i$ are bounded and continuous by assumption. Therefore, we can use Theorem E.7 and get

$$\lim_{D \to \infty} \tilde{u}_i\left(\beta_i^D, \beta_{-i}\right) = \lim_{D \to \infty} \int_{\mathcal{A}} u_i(a) dP\left(a \mid \beta_i^D, \beta_{-i}\right) = \tilde{u}_i\left(\bar{\beta}_i, \beta_{-i}\right).$$

As this holds for every $\epsilon > 0$, we get for $\Sigma_i^{SF} := \bigcup_{D \in \mathbb{N}} \Sigma_i^D$

$$\sup_{\beta'_i \in \Sigma_i^{Lip,\, p}} \tilde{u}_i\left(\beta'_i, \beta_{-i}\right) \le \sup_{\beta_i^{SF} \in \Sigma_i^{SF}} \tilde{u}_i\left(\beta_i^{SF}, \beta_{-i}\right),$$

finishing the first statement. For the second statement, note that due to the boundedness of $u_i$, we can use Kolmogorov's law of large numbers and get

$$\lim_{M_{IS} \to \infty} \hat{u}_i(\beta \mid B_{M_{IS}}) = \tilde{u}_i(\beta). \tag{24}$$

Furthermore, we get that

$$\left|\ell_i^{ver}(\beta) - \tilde{\ell}_i^{Lip,\, p}(\beta)\right| = \left|\hat{u}_i^{ver,\, D}(\beta_{-i} \mid A_{M_{IS}}) - \hat{u}_i(\beta \mid B_{M_{IS}}) - \sup_{\beta'_i \in \Sigma_i^{Lip,\, p}} \tilde{u}_i(\beta'_i, \beta_{-i}) + \tilde{u}_i(\beta)\right|$$

$$\leq \left| \hat{u}_i^{\text{ver, D}}(\beta_{-i} \,|\, A_{M_{\text{IS}}}) - \sup_{\beta_i' \in \Sigma_i^{\text{D}}} \tilde{u}_i(\beta_i', \beta_{-i}) \right| + \left| \sup_{\beta_i' \in \Sigma_i^{\text{D}}} \tilde{u}_i(\beta_i', \beta_{-i}) - \sup_{\beta_i' \in \Sigma_i^{\text{Lip, p}}} \tilde{u}_i(\beta_i', \beta_{-i}) \right|$$

$$+ |\hat{u}_i(\beta \,|\, B_{M_{\text{IS}}}) - \tilde{u}_i(\beta)|$$

$$= \varepsilon_{\text{D}} + \varepsilon_{M_{\text{IS}}} + |\hat{u}_i(\beta \,|\, B_{M_{\text{IS}}}) - \tilde{u}_i(\beta)|.$$

From the first statement and using the relation of Equation 24, we get

$$\lim_{D \to \infty} \lim_{M_{\text{IS}} \to \infty} \left| \ell_i^{\text{ver}}(\beta) - \tilde{\ell}_i^{\text{Lip, p}}(\beta) \right| \geq 0,$$

finishing the statement. $\qquad \square$

It is important to note that while some of the assumptions made in our analysis are relatively mild, others are not satisfied by some interesting settings. Let's discuss these assumptions in more detail.

Firstly, assuming the signal and action spaces to be bounded can be considered a mild restriction. Most settings already satisfy this assumption, and imposing bounds on unbounded variables usually has little practical relevance. For instance, in auctions, while bids may not have an upper bound in general, capping them to a sufficiently high value has no impact on known strategic considerations in most cases. The second assumption deals with nature being Lipschitz continuous in the players' actions, which can also be considered a weak assumption. In many settings, nature is treated as a fixed probability distribution where events are drawn independently of the players' actions. The assumption that the opponent strategies $\beta_{-i}$ are Lipschitz continuous usually is satisfied in our use cases, as many learning algorithms' parameterizations naturally satisfy this condition. However, for example, considering step functions for the opponents as well does not satisfy this assumption.

Lastly, the most stringent restrictions involve the signaling functions being Lipschitz continuous and the ex-post utilities being continuous. While relevant settings fulfill these assumptions, e.g., dynamic oligopolies (Bylka et al., 2000), several other interesting ones do not. For instance, in market settings with indivisible goods, the allocation function is discontinuous, violating both Lipschitz continuity of the signaling functions and continuity of the utilities. However, as this assumption is common for many theoretical guarantees (Glicksberg, 1952; Reny, 1999; Ui, 2016), there is ongoing research to relate the original game to a smoothed version (Kohring et al., 2023) to overcome these challenges.

Despite our statement's limitations, we conjecture that the above result does hold under less strict assumptions as well. In Supplement C.2, we observe empirically that the verifier also reliably estimates the utility loss in settings where some of the assumptions are not satisfied. We leave a theoretical analysis of this observation to future work.

# E    Auxiliary Results for the Main Theorem

**Lemma E.1.** *Let $\Gamma = (\mathcal{N}, T, S, \mathcal{A}, p, \sigma, u)$ be a multi-stage game under Theorem D.2, $\beta_{-i} \in \Sigma_{-i}$, and $\beta_i \in \Sigma_i^D$ for $D \in \mathbb{N}$. For a grid cell $C_{it}^k \subset S_{it}$ that $C_{it}^k$ is reachable under $\beta_{-i}$ and $a_{it}^k \in \mathcal{A}_{it}^D$, consider $J \subset \{1, \ldots, G_{S_{it+1}}\}$ such that $C_{it+1}^j$ is reachable from $C_{it}^k$ by taking $a_{it}^k$ under $\beta_{-i}$, i.e., $P_{it+1}\left(C_{it+1}^j | (\beta_i)^{C_{it}^k, a_{it}^k}, \beta_{-i}\right) > 0$, then there is the following relationship between the conditional probabilities from stage $t+1$ to stage $t$:*

$$P_{it}\left(C_{it}^k | (\beta_i)^{C_{it}^k, a_{it}^k}, \beta_{-i}\right) \cdot \tilde{u}_i^{c, D}\left(\beta_i, \beta_{-i} | C_{it}^k, a_{it}^k\right)$$

$$= \sum_{j \in J} P_{it+1}\left(C_{it+1}^j | (\beta_i)^{C_{it+1}^j, \beta_i(C_{it+1}^j)}, \beta_{-i}\right) \cdot \tilde{u}_i^{c, D}\left(\beta_i, \beta_{-i} | C_{it+1}^j, \beta_i(C_{it+1}^j)\right)$$

*In particular, it holds that*

$$\tilde{u}_i(\beta_i, \beta_{-i}) = \sum_{k=1}^{G_{S_{i1}}^D} P_{i1}\left(C_{i1}^k | \beta_i, \beta_{-i}\right) \cdot \tilde{u}_i^{c, D}\left(\beta_i, \beta_{-i} \,|\, C_{i1}^k, \beta_i(C_{i1}^k)\right).$$

*So, choosing $a_{it}^k = \beta_i(C_{it}^k)$ for every $t$, we can calculate player $i$'s ex-ante utility $\tilde{u}_i(\beta_i, \beta_{-i})$ by iteratively summing up the conditional probabilities.*

*Proof.* The second statement follows directly from the first by seeing that $G_{S_{i1}^D} = 1$, as there is only a single signal that can be received in stage $t = 1$.

For the first statement, note that due to construction and perfect recall, it holds that $(\beta_i)^{C_{it}^k, a_{it}^k} = (\beta_i)^{C_{it+1}^j, \beta_i(C_{it+1}^j)}$ for all $j \in J$. We then have

$$\sum_{j \in J} P_{it+1}\left(C_{it+1}^j | (\beta_i)^{C_{it+1}^j, \beta_i(C_{it+1}^j)}, \beta_{-i}\right) \cdot \tilde{u}_i^{\text{c, D}}\left(\beta_i, \beta_{-i} | C_{it+1}^j, \beta_i(C_{it+1}^j)\right) \tag{25}$$

$$= \sum_{j \in J} P_{it+1}\left(C_{it+1}^j | (\beta_i)^{C_{it+1}^j, \beta_i(C_{it+1}^j)}, \beta_{-i}\right) \cdot \int_{\mathcal{A}} u_i(a) dP\left(a | C_{it+1}^j, (\beta_i)^{C_{it+1}^j, \beta_i(C_{it+1}^j)}, \beta_{-i}\right) \tag{26}$$

$$= \sum_{j \in J} \int_{\{a \in \mathcal{A} | \sigma_{it+1}(a_{<t+1} \in C_{it+1}^j)\}} u_i(a) dP\left(a | (\beta_i)^{C_{it+1}^j, \beta_i(C_{it+1}^j)}, \beta_{-i}\right) \tag{27}$$

$$= \sum_{j \in J} \int_{\{a \in \mathcal{A} | \sigma_{it+1}(a_{<t+1} \in C_{it+1}^j)\}} u_i(a) dP\left(a | (\beta_i)^{C_{it}^k, a_{it}^k}, \beta_{-i}\right) \tag{28}$$

$$= \int_{\{a \in \mathcal{A} | \sigma_{it+1}(a_{<t+1} \in \bigcup_{j \in J} C_{it+1}^j)\}} u_i(a) dP\left(a | (\beta_i)^{C_{it}^k, a_{it}^k}, \beta_{-i}\right) \tag{29}$$

$$= P_{it}\left(C_{it}^k | (\beta_i)^{C_{it}^k, a_{it}^k}, \beta_{-i}\right) \cdot \tilde{u}_i\left((\beta_i)^{C_{it}^k, a_{it}^k}, \beta_{-i} | C_{it}^k\right) \tag{30}$$

$$= P_{it}\left(C_{it}^k | (\beta_i)^{C_{it}^k, a_{it}^k}, \beta_{-i}\right) \cdot \tilde{u}_i^{\text{c, D}}\left(\beta_i, \beta_{-i} | C_{it}^k, a_{it}^k\right), \tag{31}$$

where we used the definitions of the counterfactual conditional utilities and the conditional measures in Equations 26 and 27. Finally, we used that the $C_{it+1}^j$'s are disjoint and $P_{it}\left(C_{it}^k | (\beta_i)^{C_{it}^k, a_{it}^k}, \beta_{-i}\right) = \sum_{j \in J} P_{it+1}\left(C_{it+1}^j | (\beta_i)^{C_{it}^k, a_{it}^k}, \beta_{-i}\right)$ in Equations 29 and 30 respectively. $\square$

**Lemma E.2.** *For a given multi-stage game* $\Gamma = (\mathcal{N}, T, S, \mathcal{A}, p, \sigma, u)$*, under Theorem D.2, opponent strategies* $\beta_{-i} \in \Sigma_{-i}$*, and precision parameter* $D \in \mathbb{N}$*, it holds that*

$$\tilde{u}_i\left(\beta_i^{D,*}, \beta_{-i}\right) = \sup_{\beta_i' \in \Sigma_i^D} \tilde{u}_i\left(\beta_i', \beta_{-i}\right).$$

*Proof.* First, note the following property for conditional probabilities. For $\beta_i, \beta_i' \in \Sigma_i^D$ and any grid cell $C_{it}^k \subset S_{it}^k$, it holds that

$$P_{it}\left(C_{it}^k | (\beta_i)^{C_{it}^k, \beta_i(C_{it}^k)}, \beta_{-i}\right) = P_{it}\left(C_{it}^k | (\beta_i')^{C_{it}^k, \beta_i'(C_{it}^k)}, \beta_{-i}\right). \tag{32}$$

That is due to two reasons. First, the conditional probabilities of stage $t$ only depend on the strategies prior to stage $t$. Second, a discrete strategy $(\beta_i)^{C_{it}^k, a_{it}}$ conditioned on grid cell $C_{it}^k$ is independent of $\beta_{i<t}$.

Next, we show that for all $it \in L$, $\beta_i \in \Sigma_i^D$, and $1 \leq k \leq G_{S_{it}}$ the following holds

$$\tilde{u}_i^{\text{c, D}}\left(\beta_i, \beta_{-i} | C_{it}^k, \beta_i(C_{it}^k)\right) \leq \tilde{u}_i^{\text{c, D}}\left(\beta_i^{D,*}, \beta_{-i} | C_{it}^k, \beta_i^{D,*}(C_{it}^k)\right). \tag{33}$$

We perform a proof by induction. Let $k \in \{1, \ldots G_{S_{iT}}\}$, then it holds that

$$\tilde{u}_i^{\text{c, D}}\left(\beta_i, \beta_{-i} | C_{iT}^k, \beta_i(C_{iT}^k)\right) \leq \max_{a_{iT} \in \mathcal{A}_{iT}^D} \tilde{u}_i^{\text{c, D}}\left(\beta_i, \beta_{-i} | C_{iT}^k, a_{iT}\right)$$

$$= \tilde{u}_i^{\text{c, D}}\left(\beta_i^{D,*}, \beta_{-i} | C_{iT}^k, \beta_i^{D,*}(C_{iT}^k)\right).$$

This can be seen directly as for any $C_{it}^k$, the counterfactual conditional utility is independent of $\beta_{i<t}$. Suppose Equation 33 holds for $t + 1, \ldots, T$. Let $k \in \{1, \ldots, G_{S_{it}}\}$ and $a_{it} \in \mathcal{A}_{it}^D$. Denote with $J^{a_{it}} \subset \{1, \ldots, G_{S_{it+1}}\}$ the subset of reachable grid cells from cell $C_{it}^k$ by taking action $a_{it}$. Then we get by Theorem E.1

$$P_{it}\left(C_{it}^k | (\beta_i)^{C_{it}^k, \beta_i(C_{it}^k)}, \beta_{-i}\right) \cdot \tilde{u}_i^{\text{c, D}}\left(\beta_i, \beta_{-i} | C_{it}^k, \beta_i(C_{it}^k)\right)$$

$$= \sum_{j \in J^{\beta_i(C_{it}^k)}} P_{it+1} \left( C_{it+1}^j | (\beta_i)^{C_{it+1}^j, \beta_i(C_{it+1}^j)}, \beta_{-i} \right) \cdot \tilde{u}_i^{c, D} \left( \beta_i, \beta_{-i} | C_{it+1}^j, \beta_i(C_{it+1}^j) \right)$$

$$\leq \max_{a_{it} \in \mathcal{A}_{it}^D} \sum_{j \in J^{a_{it}}} P_{it+1} \left( C_{it+1}^j | (\beta_i)^{C_{it+1}^j, \beta_i(C_{it+1}^j)}, \beta_{-i} \right) \cdot \tilde{u}_i^{c, D} \left( \beta_i, \beta_{-i} | C_{it+1}^j, \beta_i(C_{it+1}^j) \right)$$

$$\overset{(IS)}{\leq} \max_{a_{it} \in \mathcal{A}_{it}^D} \sum_{j \in J^{a_{it}}} P_{it+1} \left( C_{it+1}^j | (\beta_i)^{C_{it+1}^j, \beta_i(C_{it+1}^j)}, \beta_{-i} \right) \cdot \tilde{u}_i^{c, D} \left( \beta_i^{D,*}, \beta_{-i} | C_{it+1}^j, \beta_i^{D,*}(C_{it+1}^j) \right)$$

$$\overset{(*)}{=} \sum_{j \in J^{\beta_i^{D,*}(C_{it}^k)}} P_{it+1} \left( C_{it+1}^j | \left( \beta_i^{D,*} \right)^{C_{it+1}^j, \beta_i^{D,*}(C_{it+1}^j)}, \beta_{-i} \right) \cdot \tilde{u}_i^{c, D} \left( \beta_i^{D,*}, \beta_{-i} | C_{it+1}^j, \beta_i^{D,*}(C_{it+1}^j) \right)$$

$$= P_{it} \left( C_{it}^k | \left( \beta_i^{D,*} \right)^{C_{it}^k, \beta_i^{D,*}(C_{it}^k)}, \beta_{-i} \right) \cdot \tilde{u}_i^{c, D} \left( \beta_i^{D,*}, \beta_{-i} | C_{it}^k, \beta_i^{D,*}(C_{it}^k) \right),$$

where (IS) denotes the induction step. Furthermore, we used Equation 32 and the definition of $\beta_{it}^{D,*}$ from Equation 19 in step (*). By applying Equation 32 again, we get the statement from Equation 33. Finally, by applying Theorem E.1, we get the statement. $\qquad\square$

**Lemma E.3.** *Let $\Gamma = (\mathcal{N}, T, S, \mathcal{A}, p, \sigma, u)$ be a multi-stage game, $\beta_{-i} \in \Sigma_{-i}$, assuming Assumptions D.2 and D.3 hold, $D \in \mathbb{N}$, and $A_{M_{IS}}$ with a number of $M_{IS} \in \mathbb{N}$ initial simulations. It holds that*

$$\lim_{M_{IS} \to \infty} \varepsilon_{M_{IS}} = 0 \text{ almost surely.}$$

*Proof.* Let $C_{it}^k \subset S_{it}$ be reachable, i.e., $P_{it}(C_{it}^k | \beta_i', \beta_{-i}) > 0$ for some $\beta_i' \in \Sigma_i^D$, and $a_{it}^k \in \mathcal{A}_{it}^D$ and $\beta_i \in \Sigma_i^D$ be arbitrary. Then it follows that $M(C_{it}^k) \to \infty$ for $M_{IS} \to \infty$. That is, for any reachable $C_{it}^k$, we sample infinitely often from the conditional counterfactual measure $P(\cdot | C_{it}^k, (\beta_i)^{C_{it}^k, a_{it}^k}, \beta_{-i})$. Furthermore, it holds that

$$\tilde{u}_i^{c, D}(\beta_i, \beta_{-i} | C_{it}^k, a_{it}^k) \leq \int_{\mathcal{A}} |u_i(a)| \, dP(a | C_{it}^k, (\beta_i)^{C_{it}^k, a_{it}^k}, \beta_{-i}) \leq ||u_i||_\infty < \infty.$$

Finally, as $a^l \in A \left( \beta_{i>t}, C_{it}^k, a_{it}^k; M_{IS} \right)$ is distributed according to $P(\cdot | C_{it}^k, (\beta_i)^{C_{it}^k, a_{it}^k}, \beta_{-i})$, Kolmogorov's law of large numbers holds and

$$\lim_{M_{IS} \to \infty} \hat{u}_i^{c, D} \left( \beta_{i>t}, \beta_{-i} | A \left( \beta_{i>t}, C_{it}^k, a_{it}^k; M_{IS} \right) \right) = \tilde{u}_i^{c, D} \left( \beta_i, \beta_{-i} | C_{it}^k, a_{it}^k \right) \text{ almost surely.}$$

In particular, it holds that

$$\lim_{M_{IS} \to \infty} \hat{u}_i^{c, D} \left( \beta_{-i} | A \left( C_{iT}^k, \beta_i^{D, M_{IS}}(C_{iT}^k); M_{IS} \right) \right) = \tilde{u}_i^{c, D} \left( \beta_i, \beta_{-i} | C_{iT}^k, \beta_i^{D,*}(C_{iT}^k) \right) \text{ almost surely,}$$

as the utility is independent from the choice of the $\arg\max$ in Equation 18. Therefore, following the backward induction procedure from Equation 19, we get

$$\lim_{M_{IS} \to \infty} \hat{u}_i^{c, D} \left( \beta_{i>t}^{D, M_{IS}}, \beta_{-i} | A \left( \beta_{i>t}^{D, M_{IS}}, C_{it}^k, \beta_{i>t}^{D, M_{IS}}(C_{it}^k); M_{IS} \right) \right) = \tilde{u}_i^{c, D} \left( \beta_i^{D,*}, \beta_{-i} | C_{it}^k, \beta_i^{D,*}(C_{it}^k) \right) \text{ a. s.}$$

Finally, we get

$$\lim_{M_{IS} \to \infty} \hat{u}_i^{\text{ver}, D}(\beta_{-i} | A_{M_{IS}}) = \tilde{u}_i \left( \beta_i^{D,*}, \beta_{-i} \right) = \sup_{\beta_i' \in \Sigma_i^D} \tilde{u}_i \left( \beta_i', \beta_{-i} \right) \text{ almost surely,}$$

where we used Theorem E.1 for the first, and Theorem E.2 for the second equation. This gives us the desired statement. $\quad\square$

**Lemma E.4.** *Let $\Gamma = (\mathcal{N}, T, S, \mathcal{A}, p, \sigma, u)$ be a multi-stage game, where Theorem D.2 holds. Let $\mathcal{G}_{it} = (S_{it}^D, \mathcal{A}_{it}^D, C_{it}, D)$ be the discretization that results from a regular grid of $2^D$ points along each dimension of $S_{it}$ and $\mathcal{A}_{it}$. Then for every pure Lipschitz continuous strategy $\beta_{it} \in \Sigma_{it}^{Lip, p}$ there exists a sequence $\{\beta_{it}^D\}_{D \in \mathbb{N}}$ with $\beta_{it}^D \in \Sigma_{it}^D(\mathcal{G}_{it})$ such that*

$$\lim_{D \to \infty} ||\beta_{it} - \beta_{it}^D||_\infty = 0.$$

*Proof.* Let $\beta_{it} \in \Sigma_{it}^{\text{Lip, p}}$. Then there exists an $L > 0$ such that for $s, s' \in S_{it}$ it holds that

$$d_W\left(\beta_{it}(s), \beta_{it}(s')\right) = ||\beta_{it}(s) - \beta_{it}(s')||_\infty \leq L \cdot ||s - s'||.$$

As $\mathcal{A}_{it}$ and $S_{it}$ are compact, there exists a $K > 0$ such that $||a - a'||_\infty \leq K$ and $||s - s'||_\infty \leq K$ for all $a, a' \in \mathcal{A}_{it}$ and $s, s' \in S_{it}$. Then, for every $a \in \mathcal{A}_{it}$ and $s \in S_{it}$, there exist $\tilde{a} \in \mathcal{A}_{it}^{\text{D}}$ and $\tilde{s} \in S_{it}^{\text{D}}$ such that $||a - \tilde{a}||_\infty \leq K2^{-\text{D}}$ and $||s - \tilde{s}||_\infty \leq K2^{-\text{D}}$.

Remember that $\left(C_{it}^k\right)_{1 \leq k \leq G_{S_{it}^{\text{D}}}}$ partitions $S_{it}$. Let $s \in C_{it}^k$, then there is a unique $s^k \in S_{it}^{\text{D}} \cap C_{it}^k$. Define

$$\beta_{it}^{\text{D}}(s) = \underset{a_{it}^{Dk} \in \mathcal{A}_{it}^D}{\arg\min} \left||a_{it}^k - \beta_{it}(s^k)\right||_\infty \text{ for } s \in C_{it}^k, 1 \leq k \leq G_{S_{it}}^{\text{D}}.$$

Then $\beta_{it}^{\text{D}} \in \Sigma_{it}^{\text{D}}$ for every $\text{D} \in \mathbb{N}$. Finally, we get for all $s \in S_{it}$ that

$$\left||\beta_{it}(s) - \beta_{it}^{\text{D}}(s^k)\right||_\infty \leq \left||\beta_{it}(s) - \beta_{it}(s^k)\right||_\infty + \left||\beta_{it}(s^k) - \beta_{it}^{\text{D}}(s^k)\right||_\infty$$
$$\leq L\left||s - s^k\right|| + K2^{-\text{D}} \leq K \cdot (1 + L)2^{-\text{D}}.$$

As $K$ and $L$ are constants, and $2^{-\text{D}} \to 0$ for $\text{D} \to \infty$, we get the statement. $\qquad\square$

We now turn to translate a close approximation of a strategy $\beta_i$ by a step function $\beta_i'$ into closeness of the outcome distribution in the Wasserstein distance. For completeness, we restate some well-known results about the Wasserstein distance.

**Definition E.5** (Wasserstein distance). (Villani, 2009, p.93) Let $(X, d)$ be a Polish metric space. For any probability measures $\mu, \nu$ on $X$, the (1-)Wasserstein distance between $\mu$ and $\nu$ is defined by

$$d_W(\mu, \nu) = \inf_{\pi \in \Pi(\mu, \nu)} \int_X d(x, y) d\pi(x, y),$$

where $\Pi(\mu, \nu)$ denotes the space of couplings between $\mu$ and $\nu$. That is $\pi \in \Pi(\mu, \nu)$ is a probability measure on $X \times X$, such that $\int_X \pi(x, y) dy = \mu(x)$ and $\int_X \pi(x, y) dx = \nu(x)$.

In our applications, we always assume the metric $d$ to be induced by some norm $||\cdot||$. As we consider finite-dimensional spaces by Assumption D.2, the choice of a norm is irrelevant.

**Lemma E.6** (Kantorovich–Rubinstein duality). *(Villani, 2009, p.59) Let $(X, d)$ be a Polish metric space. For any probability measures $\mu, \nu$ on $X$, and $K > 0$, there holds the following equality*

$$d_W(\mu, \nu) = \frac{1}{K} \sup_{||f||_{Lip} \leq K} \left\{ \int f d\mu - \int f d\nu \right\},$$

*where $||\cdot||_{Lip}$ denotes the Lipschitz norm.*

**Theorem E.7** (Metrization of weak convergence). *(Villani, 2009, p.96) Let $(X, d)$ be a Polish metric space and $(\mu_k)_{k \in \mathcal{N}}$ is a sequence of measures in $P(X)$, and $\mu \in P(X)$, then the following two statements are equivalent*

1. $d_W(\mu_k, \mu) \to 0$

2. *For all bounded continuous functions $f : X \to \mathbb{R}$, one has*

$$\int f d\mu_k \to \int f d\mu.$$

To proof Theorem D.4, we leverage Theorem E.7. As the utilities are assumed to be continuous, it suffices to show that closeness to a Lipschitz continuous strategy translates to a small Wasserstein distance of the outcome distribution. More specifically, we show that under Assumptions D.2 and D.3, for every $\beta_i \in \Sigma_i^{\text{Lip, p}}$ there exists a sequence $\left(\beta_i^{\text{D}}\right)_{\text{D} \in \mathcal{N}}$ with $\beta_i^{\text{D}} \in \Sigma_i^{\text{D}}$ such that $d_W\left(P\left(\cdot \mid \beta_i, \beta_{-i}\right), P\left(\cdot \mid \beta_i^{\text{D}}, \beta_{-i}\right)\right) \to 0$. For this, we show some intermediate results first.

**Lemma E.8.** *Let $\mu_1, \nu_1$ and $\mu_2, \nu_2$ be measures on $\mathbb{R}^n$ and $\mathbb{R}^m$ respectively. Define the product measures $\mu := \mu_1 \otimes \mu_2$, $\nu := \nu_1 \otimes \nu_2$. Then the following inequality holds*

$$d_W(\mu, \nu) \leq d_W(\mu_1, \nu_1) + d_W(\mu_2, \nu_2)$$

*Proof.* By Theorem 4.1 of (Villani, 2009, p.43), there exist optimal couplings $\pi_1, \pi_2$ for $\mu_1, \nu_1$ and $\mu_2, \nu_2$ respectively, such that

$$d_W(\mu_1, \nu_1) = \int_{\mathbb{R}^n \times \mathbb{R}^n} d_{\mathbb{R}^n}(x_1, x_2) d\pi_1(x_1, x_2) , \, d_W(\mu_2, \nu_2) = \int_{\mathbb{R}^m \times \mathbb{R}^m} d_{\mathbb{R}^m}(y_1, y_2) d\pi_2(y_1, y_2).$$

Then $\pi := \pi_1 \otimes \pi_2$ is the trivial coupling for $\mu$ and $\nu$, which can be readily checked by

$$\int_{\mathbb{R}^n \times \mathbb{R}^m} \pi(x_1, x_2, y_1, y_2) d(x_1, y_1) = \int_{\mathbb{R}^n} \pi_1(x_1, x_2) dx_1 \int_{\mathbb{R}^m} \pi_2(y_1, y_2) dy_1 = \nu_1(x_2)\nu_2(y_2),$$

$$\int_{\mathbb{R}^n \times \mathbb{R}^m} \pi(x_1, x_2, y_1, y_2) d(x_2, y_2) = \int_{\mathbb{R}^n} \pi_1(x_1, x_2) dx_2 \int_{\mathbb{R}^m} \pi_2(y_1, y_2) dy_2 = \mu_1(x_1)\mu_2(y_1).$$

Therefore, we get

$$\begin{aligned}
d_W(\mu, \nu) &\leq \int_{\mathbb{R}^{n+m} \times \mathbb{R}^{n+m}} d_{\mathbb{R}^{n+m}}(x_1, y_1, x_2, y_2) d\pi(x_1, x_2, y_1, y_2) \\
&\leq \int_{\mathbb{R}^{n+m} \times \mathbb{R}^{n+m}} d_{\mathbb{R}^n}(x_1, y_1) + d_{\mathbb{R}^m}(x_2, y_2) d\pi(x_1, x_2, y_1, y_2) \\
&= \int_{\mathbb{R}^n \times \mathbb{R}^n} d_{\mathbb{R}^n}(x_1, y_1) d\pi_1(x_1, y_1) + \int_{\mathbb{R}^m \times \mathbb{R}^m} d_{\mathbb{R}^m}(x_2, y_2) d\pi_2(x_2, y_2) \\
&= d_W(\mu_1, \nu_1) + d_W(\mu_2, \nu_2)
\end{aligned}$$

$\square$

**Lemma E.9.** *Let $\beta_{it}, \beta'_{it} \in \Sigma^p_{it}$, $\beta_{-it} \in \Sigma_{-it}$ and $\beta_{<t} \in \Sigma_{<t}$. Under Assumption D.2, it holds that*

$$d_W\left(P_{<t+1}\left(\cdot \mid (\beta_{i<t}, \beta_{it}), (\beta_{-i<t}, \beta_{-it})\right), P_{<t+1}\left(\cdot \mid (\beta_{i<t}, \beta'_{it}), (\beta_{-i<t}, \beta_{-it})\right)\right) \leq ||\beta_{it} - \beta'_{it}||_\infty.$$

*Proof.* We start with showing the following step first

$$d_W\left(P_t\left(\cdot \mid a_{<t}, \beta_{it}, \beta_{-it}\right), P_t\left(\cdot \mid a_{<t}, \beta'_{it}, \beta_{-it}\right)\right) \leq ||\beta_{it} - \beta'_{it}||_\infty \text{ for all } a_{<t} \in \mathcal{A}_{<t}. \tag{34}$$

Let $a_{<t} \in \mathcal{A}_{<t}$ be arbitrary. Then note first that as $\beta_{it}, \beta'_{it}$ are pure strategies, $\beta_{it}(\cdot \mid \sigma_{it}(a_{<t}))$ and $\beta'_{it}(\cdot \mid \sigma_{it}(a_{<t}))$ are Dirac-measures. Therefore, we can use the well-known fact that the Wasserstein distance between these is simply the distance between the points with positive measure (see Example 6.3 in (Villani, 2009, p.94)), i.e.,

$$d_W\left(\beta_{it}(\cdot \mid \sigma_{it}(a_{<t})), \beta'_{it}(\cdot \mid \sigma_{it}(a_{<t}))\right) = ||\beta_{it}(\sigma_{it}(a_{<t})) - \beta'_{it}(\sigma_{it}(a_{<t}))||_\infty \leq ||\beta_{it} - \beta'_{it}||_\infty ,$$

where we abused notation and treated $\beta_{it}, \beta'_{it}$ once as mapping to Dirac-measures and once as mapping to elements in $\mathcal{A}_{it}$. By Theorem D.2, $P_t$ is a product measure on $\mathbb{R}^m$ for some $m \in \mathbb{N}$. Therefore, one can use Theorem E.8 and get

$$\begin{aligned}
&d_W\left(P_t\left(\cdot \mid a_{<t}, \beta_{it}, \beta_{-it}\right), P_t\left(\cdot \mid a_{<t}, \beta'_{it}, \beta_{-it}\right)\right) \\
&\leq d_W\left(\beta_{it}(\cdot \mid \sigma_{it}(a_{<t})), \beta'_{it}(\cdot \mid \sigma_{it}(a_{<t}))\right) \leq ||\beta_{it} - \beta'_{it}||_\infty ,
\end{aligned}$$

which shows Equation 34. Consequently, we get

$$\begin{aligned}
&d_W\left(P_{<t+1}\left(\cdot \mid (\beta_{i<t}, \beta_{it}), (\beta_{-i<t}, \beta_{-it})\right), P_{<t+1}\left(\cdot \mid (\beta_{i<t}, \beta'_{it}), (\beta_{-i<t}, \beta_{-it})\right)\right) \\
&= \sup_{||f||_{\text{Lip}} \leq 1} \int_{\mathcal{A}_{<t}} \int_{\mathcal{A}_{\cdot t}} f(a_{<t}, a_{\cdot t}) \, dP_t(a_{\cdot t} \mid a_{<t}, \beta_{it}, \beta_{-it}) \\
&\quad - \int_{\mathcal{A}_{\cdot t}} f(a_{<t}, a_{\cdot t}) \, dP_t(a_{\cdot t} \mid a_{<t}, \beta'_{it}, \beta_{-it}) \, dP_{<t}(a_{<t} \mid \beta_{<t})
\end{aligned}$$

$$\overset{(E.6)}{\leq} \int_{\mathcal{A}_{<t}} d_W \left( P_t \left( \cdot \mid a_{<t}, \beta_{it}, \beta_{-it} \right), P_t \left( \cdot \mid a_{<t}, \beta'_{it}, \beta_{-it} \right) \right) dP_{<t} \left( a_{<t} \mid \beta_{<t} \right)$$

$$\overset{Equ.(34)}{\leq} \int_{\mathcal{A}_{<t}} ||\beta_{it} - \beta'_{it}||_\infty dP_{<t} \left( a_{<t} \mid \beta_{<t} \right) = ||\beta_{it} - \beta'_{it}||_\infty.$$

$\square$

**Lemma E.10.** *Let* $(X, d_X), (Y, d_Y)$ *be metric Polish spaces,* $f : X \times Y \to \mathbb{R}$ *be a* $K_f$-*Lipschitz continuous function and* $\mu(\cdot \mid x)$ *be a measure on* $Y$ *for every* $x \in X$. *Furthermore, the mapping* $x \mapsto \mu(\cdot \mid x)$ *is* $K_\mu$-*Lipschitz continuous with respect to the Wasserstein distance* $d_W$. *Then it holds that*

$$g_f : X \to \mathbb{R}, x \mapsto \int_Y f(x, y) d\mu(y|x) \text{ is } (K_f + K_f K_\mu) - \text{Lipschitz.}$$

*Proof.* Let $x, x' \in X$, then

$$|g_f(x) - g_f(x')| = \left| \int_Y f(x, y) d\mu(y|x) - \int_Y f(x', y) d\mu(y|x') \right|$$

$$\leq \left| \int_Y f(x, y) - f(x', y) d\mu(y|x) \right| + \left| \int_Y f(x', y) d\mu(y|x) - \int_Y f(x', y) d\mu(y|x') \right|$$

$$\leq \int_Y K_f \cdot d_{X \times Y} \left( (x, y)^T, (x', y)^T \right) d\mu(y|x) + \sup_{||g||_{\text{Lip}} \leq K_f} \left| \int_Y g(y) d\mu(y|x) - \int_Y g(y) d\mu(y|x') \right|$$

$$\overset{(E.6)}{=} K_f \int_Y d_X(x, x') d\mu(y|x) + K_f \cdot d_W \left( \mu(\cdot \mid x), \mu(\cdot \mid x') \right)$$

$$= K_f \left( d_X(x, x') + d_W \left( \mu(\cdot \mid x), \mu(\cdot \mid x') \right) \right)$$

$$\leq K_f \left( d_X(x, x') + L_\mu d_X(x, x') \right) = (K_f + K_f K_\mu) d_X(x, x').$$

$\square$

**Lemma E.11.** *Let Assumptions D.2 and D.3 hold, and let* $\beta_{<t}, \beta'_{<t} \in \Sigma_{<t}$ *and* $\beta_{\cdot t} \in \Sigma^{Lip}_{\cdot t}$. *Then, there exists* $K > 0$ *such that*

$$d_W \left( P_{<t+1} \left( \cdot \mid \beta_{<t}, \beta_{\cdot t} \right), P_{<t+1} \left( \cdot \mid \beta'_{<t}, \beta_{\cdot t} \right) \right) \leq K \cdot d_W \left( P_{<t} \left( \cdot \mid \beta_{<t} \right), P_{<t} \left( \cdot \mid \beta'_{<t} \right) \right)$$

*Proof.* By Theorem D.3, there exist constants $K_{\sigma_{it}} > 0$ for $it \in L$, such that $\sigma_{it}$ is $K_{\sigma_{it}}$-Lipschitz continuous in $\mathcal{A}_{<t}$. Also, denote with $K_{0t}$ nature's Lipschitz constant with respect to $d_W$ in stage $t$. Similarly, as $\beta_{\cdot t} \in \Sigma^{Lip}_{\cdot t}$, there exist constants $K_{\beta_{it}} > 0$ such that $\beta_{it}(\cdot \mid s_{it})$ is $K_{\beta_{it}}$-Lipschitz with respect to the Wasserstein distance. Overall, we get for $it \in L$, $a_{<t}, a'_{<t} \in \mathcal{A}_{<t}$

$$d_W \left( \beta_{it} \left( \cdot \mid \sigma_{it}(a_{<t}) \right), \beta_{it} \left( \cdot \mid \sigma_{it}(a'_{<t}) \right) \right) \leq K_{\beta_{it}} K_{\sigma_{it}} \cdot ||a_{<t} - a'_{<t}||.$$

Denote $K_t := K_{0t} + \sum_{i \in \mathcal{N}} K_{\beta_{it}} K_{\sigma_{it}}$. Then we get that the mapping $a_{<t} \mapsto P_t \left( \cdot \mid a_{<t}, \beta_{\cdot t} \right)$ is $K_t$-Lipschitz continuous with respect to $d_W$, which can be seen by

$$d_W \left( P_t \left( \cdot \mid a_{<t}, \beta_{\cdot t} \right), P_t \left( \cdot \mid a'_{<t}, \beta_{\cdot t} \right) \right)$$

$$\overset{(E.8)}{\leq} d_W \left( p_{0t} \left( \cdot \mid a_{<t} \right), p_{0t} \left( \cdot \mid a'_{<t} \right) \right) + \sum_{i \in \mathcal{N}} d_W \left( \beta_{it} \left( \cdot \mid \sigma_{it}(a_{<t}) \right), \beta_{it} \left( \cdot \mid \sigma_{it}(a'_{<t}) \right) \right)$$

$$\leq \left( K_{0t} + \sum_{i \in \mathcal{N}} K_{\beta_{it}} K_{\sigma_{it}} \right) \cdot ||a_{<t} - a'_{<t}||_\infty,$$

for all $a_{<t}, a'_{<t} \in \mathcal{A}_{<t}$. Let $f : \mathcal{A}_{<t+1} \to \mathbb{R}$ be 1-Lipschitz continuous. Then, we get by Lemma E.10, that the function $g_f(a_{<t}) := \int_{\mathcal{A}_{<t}} f(a_{<t}, a_{\cdot t}) dP_t \left( a_{\cdot t} \mid a_{<t}, \beta_{\cdot t} \right)$ is $(1 + K_t)$-Lipschitz continuous. With this, we get

$$d_W \left( P_{<t+1} \left( \cdot \mid \beta_{<t}, \beta_{\cdot t} \right), P_{<t+1} \left( \cdot \mid \beta'_{<t}, \beta_{\cdot t} \right) \right)$$

$$\overset{(E.6)}{=} \sup_{||f||_{\text{Lip}} \leq 1} \int_{\mathcal{A}_{<t+1}} f(a_{<t+1}) P_{<t+1} \left( a_{<t+1} \mid \beta_{<t}, \beta_{\cdot t} \right) - \int_{\mathcal{A}_{<t+1}} f(a_{<t+1}) P_{<t+1} \left( a_{<t+1} \mid \beta'_{<t}, \beta_{\cdot t} \right)$$

$$= \sup_{||f||_{\text{Lip}} \leq 1} \int_{\mathcal{A}_{<t}} g_f(a_{<t}) P_{<t} \left( a_{<t} \mid \beta_{<t} \right) - \int_{\mathcal{A}_{<t}} g_f(a_{<t}) P_{<t} \left( a_{<t} \mid \beta'_{<t} \right)$$

$$\overset{(E.10)}{\leq} \sup_{||g||_{\text{Lip}} \leq 1 + K_t} \int_{\mathcal{A}_{<t}} g(a_{<t}) P_{<t} \left( a_{<t} \mid \beta_{<t} \right) - \int_{\mathcal{A}_{<t}} g(a_{<t}) P_{<t} \left( a_{<t} \mid \beta'_{<t} \right)$$

$$\overset{(E.6)}{=} (1 + K_t) \cdot d_W \left( P_{<t} \left( \cdot \mid \beta_{<t} \right), P_{<t} \left( \cdot \mid \beta'_{<t} \right) \right),$$

which shows the statement. $\qquad\square$

**Lemma E.12.** *Let $\Gamma = (\mathcal{N}, T, S, \mathcal{A}, p, \sigma, u)$ be a multi-stage game, where Assumptions D.2 and D.3 hold. For strategies $\beta_{-i} \in \Sigma_{-i}^{Lip}$, $\beta_i \in \Sigma_i^{Lip,\, p}$ and $\epsilon > 0$, there exists a $\delta > 0$ such that for all $\beta'_i \in \Sigma_i^p$ with $||\beta_i - \beta'_i||_\infty < \delta$, it holds that*

$$d_W \left( P \left( \cdot \mid \beta_i, \beta_{-i} \right), P \left( \cdot \mid \beta'_i, \beta_{-i} \right) \right) < \epsilon.$$

*Proof.* Let $\epsilon > 0$, $\beta_{-i} \in \Sigma_{-i}^{\text{Lip}}$, $\beta_i \in \Sigma_i^{\text{Lip},\, \text{p}}$, and $\beta'_i \in \Sigma_i^{\text{p}}$. Then we get

$$d_W \left( P \left( \cdot \mid \beta_i, \beta_{-i} \right), P \left( \cdot \mid \beta'_i, \beta_{-i} \right) \right)$$

$$\overset{(\triangle\text{-inequ.})}{\leq} \sum_{t=1}^{T} d_W \left( P \left( \cdot \mid (\beta'_{i<t}, \beta_{it}, \beta_{i>t}), \beta_{-i} \right), P \left( \cdot \mid (\beta'_{i<t}, \beta'_{it}, \beta_{i>t}), \beta_{-i} \right) \right)$$

$$\overset{(E.11)}{\leq} \sum_{t=1}^{T} \left( \prod_{v>t} K_v \right) \cdot d_W \left( P_{<t+1} \left( \cdot \mid (\beta'_{i<t}, \beta_{it}), \beta_{-i} \right), P_{<t+1} \left( \cdot \mid (\beta'_{i<t}, \beta'_{it}), \beta_{-i} \right) \right)$$

$$\overset{(E.9)}{\leq} \sum_{t=1}^{T} K_{>t} ||\beta_{it} - \beta'_{it}||_\infty,$$

where $K_{>t} = \prod_{v=t+1}^{T} K_v$ with $K_t := K_{0t} + \sum_{i \in \mathcal{N}} K_{\beta_{it}} K_{\sigma_{it}}$ (see proof of Theorem E.11) for all $1 \leq t \leq T$. By choosing $\delta < \max_{t \in T} \frac{\epsilon}{K_{>t} \cdot T}$, we get the statement. $\qquad\square$

