# OpenReview forum: "Deep Reinforcement Learning for Equilibrium Computation in Multi-Stage Auctions and Contests"
_ICML.cc/2024/Workshop/Agentic_Markets — Agentic Markets @ ICML'24 Poster_

### Official Review · Reviewer_2H26 · 2024-06-13
**An interesting and novel work in DRL and equilibrium finding.**

**Rating:** 8
**Confidence:** 3

**Review:**

The authors employ Deep Reinforcement Learning (DRL) techniques to find an equilibrium strategy in continuous multi-stage games. In the proposed method, the players update the parameters of their strategies, which are functionals, using a first-order gradient method. The authors define an appropriate metric to measure the distance of the strategy profile from the game's equilibrium and show that, under assumptions, they can estimate its value using an appropriate verifier. Finally, by taking advantage of the verifier, they provide experimental evidence that suggests that the aforementioned self-play leads, surprisingly, to the game's equilibrium.

Strengths:

1. The results are surprising and significant and are likely to inspire further research on the topic.
2. The paper is well-written.

Weaknesses:

1. The experimental results are surprising given that finding NE approximations is NP-hard. Are there any simple cases where the method does not find the NE? It would be helpful to discuss the expected limitations of the method.
2. The main body of the paper is not entirely self-contained, and requires some back-and-forth with the Appendix, e.g., what is $\tilde u$ and $L$ in Section 2?

Questions:

1. Does Theorem 4.1 assume the existence and uniqueness of an equilibrium?
2. Are there any assumptions about the learning rate of the players? I am wondering if the self-play can stuck in a recurrent set, instead of reaching the equilibrium.

---

### Official Review · Reviewer_mfYE · 2024-06-17
**Submission 11: "Deep Reinforcement Learning for Equilibrium Computation in Multi-Stage Auctions and Contests"**

**Rating:** 7
**Confidence:** 4

**Review:**

This work aims to 1) approximate equilibrium strategies in multi-stage games with continuous observation and actions spaces using deep reinforcement learning, and 2) verify exploitability (nearness to Nash) of a strategy profile using . The authors simulate and evaluate REINFORCE and PPO in a sequential sales auction setting showing they are able to closely recover the Nash equilibrium solution.

- In equation (3), shouldn't the measure $dP_{it}(\cdot \vert \beta^*)$ be inside the large parentheses? Also, if one is to approximate the integral with Monte-Carlo and then take the square root, this could lead to an underestimate. Let $X = \frac{1}{N} \sum_{j=1}^N (\beta(s_j) - \beta^*(s_j))^2$ with $s_j \sim P$ be your unbiased MC estimator of $\int (\beta - \beta^*)^2 dP$. Then Jensen's inequality says $\mathbb{E}[\sqrt{X}] \le \sqrt{\mathbb{E}[X]}$, which means that if you repeated your Table 1 experiment to estimate $L^2$ distance many times and took the average, your result would be an underestimate of the true $L^2$ distance. In summary, it might be better just to present the squared distance which you can give an unbiased estimate for.
- What are the scale of the utilities in the experiments? It would be helpful to have some reference point to understand how small the "utility losses" in Table 1 are.
- Previous work (https://www.nature.com/articles/s41598-022-20234-3) demonstrated REINFORCE (Figure 9) struggling to match equilibrium play for certain single-stage auctions; in contrast, fictious play (Figure 8) was able to recover Nash. Is it possible that the auction settings you empirically study are "easy" for REINFORCE/PPO to solve? Are there any "realistic" instances you can find where they fail?